# DeepBacs for multi-task bacterial image analysis using open-source deep learning approaches

Christoph Spahn [1,2 ✉], Estibaliz Gómez-de-Mariscal[3], Romain F. Laine [4,5,11], Pedro M. Pereira [6], Lucas von Chamier[4], Mia Conduit[7], Mariana G. Pinho [6], Guillaume Jacquemet[8,9,10], Séamus Holden[7], Mike Heilemann [2 ✉] & Ricardo Henriques [3,4,5 ✉]

This work demonstrates and guides how to use a range of state-of-the-art artificial neural-networks to analyse bacterial microscopy images using the recently developed ZeroCostDL4Mic platform. We generated a database of image datasets used to train networks for various image analysis tasks and present strategies for data acquisition and curation, as well as model training. We showcase different deep learning (DL) approaches for segmenting bright field and fluorescence images of different bacterial species, use object detection to classify different growth stages in time-lapse imaging data, and carry out DL-assisted phenotypic profiling of antibiotic-treated cells. To also demonstrate the ability of DL to enhance low-phototoxicity live-cell microscopy, we showcase how image denoising can allow researchers to attain high-fidelity data in faster and longer imaging. Finally, artificial labelling of cell membranes and predictions of super-resolution images allow for accurate mapping of cell shape and intracellular targets. Our purposefully-built database of training and testing data aids in novice users' training, enabling them to quickly explore how to analyse their data through DL. We hope this lays a fertile ground for the efficient application of DL in microbiology and fosters the creation of tools for bacterial cell biology and antibiotic research.

[1] Department of Natural Products in Organismic Interaction, Max Planck Institute for Terrestrial Microbiology, Marburg, Germany. [2] Institute of Physical and Theoretical Chemistry, Goethe-University Frankfurt, Frankfurt, Germany. [3] Instituto Gulbenkian de Ciência, 2780-156 Oeiras, Portugal. [4] MRC-Laboratory for Molecular Cell Biology, University College London, London, UK. [5] The Francis Crick Institute, London, UK. [6] Instituto de Tecnologia Química e Biológica António Xavier, Universidade Nova de Lisboa, Oeiras, Portugal. [7] Centre for Bacterial Cell Biology, Newcastle University Biosciences Institute, Faculty of Medical Sciences, Newcastle upon Tyne NE24AX, United Kingdom. [8] Turku Bioscience Centre, University of Turku and Åbo Akademi University, Turku, Finland. [9] Faculty of Science and Engineering, Cell Biology, Åbo Akademi University, Turku, Finland. [10] Turku Bioimaging, University of Turku and Åbo Akademi University, Turku, Finland. [11] Present address: Micrographia Bio, Translation and Innovation hub 84 Wood lane, W120BZ London, UK. ✉email: christoph.spahn@mpi-marburg.mpg.de; heilemann@chemie.uni-frankfurt.de; rjhenriques@igc.gulbenkian.pt

The study of microorganisms and microbial communities is a multidisciplinary approach bringing together molecular biology, biochemistry, and biophysics. It covers large spatial scales ranging from single molecules over individual cells to entire ecosystems. The amount of data collected in microbial studies constantly increases with technical developments, which can become challenging for classical data analysis and interpretation, requiring more complex computational approaches to extract relevant features from the data landscape. Therefore, manual analysis is increasingly replaced by automated analysis, particularly with machine learning (ML)[1]. In bioimage analysis, ML for example contributed to a better understanding of viral organisation[2] and the mode of action of antimicrobial compounds[3]. In recent years, the interest in ML, and particularly deep learning (DL), for bioimage analysis has increased significantly, as their high versatility allows them to perform many different image analysis tasks with high performance and speed[4–7]. This was impressively demonstrated for image segmentation[8–11], object detection and classification[12,13], quality enhancement and denoising[14,15], and even prediction of super-resolution images[14,16,17] from diffraction-limited images. DL tools have even contributed to the conception of new transformative applications such as image-to-image translation[18] or artificial labelling[19,20].

Next to developing novel DL approaches, effort has been put into their democratisation and providing an entry point for non-experts by simplifying their use and providing pretrained models[21–24]. To further democratise expensive model training, recent developments employ cloud-based hardware solutions, thus bypassing the need for specialised hardware[22,25,26]. However, these methodologies dominantly focused on the study of eukaryotes, particularly given the wealth of pre-existing imaging data[17,27].

In microbiology, DL approaches are intensively used for segmentation, as they facilitate single-cell analysis in image analysis pipelines and automated analysis of large datasets[28–33]. Such pipelines can, for example, be used for automated cell counting or morphological analysis of individual cells or cell lineages. However, due to the considerable variety in microscopy techniques and bacterial shapes there is no universal DL network that excels for all types of data. Although there is effort in developing generalist networks[10,28], specialised networks tend to especially excel for the type of images they were developed for and are typically not characterised beyond that.

Beyond segmentation, DL remains largely underexploited in the analysis of microbial bioimages, although other tasks like object detection, denoising, artificial labelling, or resolution enhancement could be well applied with many useful applications.

Object detection is a task closely related to image segmentation, which, instead of classifying pixels as background or foreground pixels, outputs a bounding box and a class label for each detected object. This is used extensively in real-life applications, such as self-driving cars or detecting items in photographs[12,13]. In microscopy, object detection can detect and classify cells of specific types or states[22], which can also be integrated into smart imaging approaches that allow for automated image acquisition[34].

Denoising and artificial labelling are particularly suited to improve live-cell imaging, where high contrast and fast image acquisition are critical to capture the dynamic nature of biology in full detail. However, these usually come associated with high illumination power regimes often not compatible with live-cell imaging[14]. Several denoising techniques such as PureDenoise[35] or DL-based approaches, both self-supervised (e.g. Noise2Void[15]) and fully supervised (e.g. Content-aware image restoration [CARE][14]), have been proposed to circumvent this experimental challenge. Robust denoising of images with low signal-to-noise ratio (SNR) allows for lower light doses, which reduces phototoxicity, and shorter integration times, which increases the temporal resolution, as was demonstrated in many eukaryotic systems[14].

Phototoxicity can be further reduced by employing artificial labelling networks. These networks create pseudo-fluorescent images from bright field, histology or electron microscopy (EM) images[19,20,36]. Artificial labelling is particularly beneficial for bright field-to-fluorescence transformation in live-cell application. As it does not require fluorophore excitation, it is even less phototoxic than denoising of low-SNR images, while providing molecular specificity. Here, the neural network learns features in bright field images imprinted by specific structures or biomolecules (for example, membranes or nucleic acids) and creates a virtual fluorescence image of these structures. In contrast to image segmentation, artificial labelling does not require manual annotation, which reduces the time required for data curation. The use of bright field images as inputs further allows to train networks for different subcellular structures, leading to a high multiplexing capability. In the original published work, this allowed predicting multiple subcellular structures and their dynamics in mammalian tissue culture samples[19,20].

Another strategy to increase the information content in microscopy images is resolution enhancement. Several supervised DL approaches were developed that allow conversion of low-resolution to high-resolution images. This includes confocal-to-STED[16], widefield-to-SRRF[14,22] or widefield-to-SIM[17] transformation. Next to increasing spatial resolution, these networks also reduce the required light dose and increase temporal resolution. This, for example, allowed for fast multi-colour imaging of organelles in live mammalian cells[17] or computational high-resolution imaging of cytoskeletal proteins or the endocytosis machinery. Application in microbiology, however, is still lacking.

To diversify the use of DL technology in microbiological application, we propose that existing open-source DL approaches used for eukaryotic specimens can be easily employed to analyse bacterial bioimages. As the key requirement for successful application of DL is suitable training data, we generated various image datasets comprising different bacterial species (Escherichia coli (E. coli), Staphylococcus aureus (S. aureus) and Bacillus subtilis (B. subtilis)) and imaging modalities (bright field, widefield and confocal fluorescence and super-resolution microscopy). We used these datasets to train DL models for a wide range of applications using the recently developed ZeroCostDL4Mic platform[22]. Due to the ease-of-use and low-cost capabilities of ZeroCostDL4Mic, it allows users to quickly train various networks and explore whether they are suitable for the desired task. DeepBacs guides users by providing data and models, as well as advice and image annotation/analysis strategies.

Specifically, we demonstrate the potential of open-source DL technology in image segmentation of both rod and spherically shaped bacteria (fluorescence and bright field images); in the detection of cells and their classification based on growth stage and antibiotic-induced phenotypic alterations; on denoising of live-cell microscopy data, such as nucleoid, MreB and FtsZ dynamics; and finally, we explore the potential of DL approaches for artificial labelling of bacterial membranes in bright field images and prediction of super-resolution images from diffraction-limited widefield images.

To give researchers the opportunity to test and explore the different DL networks, we openly share our data and models. This will enable them to use pretrained models on existing data, or to train custom models more efficiently via transfer learning[22,37]. We envision that this work will help microbiologists seamlessly leverage DL for microscopy image analysis and benefit from high-performance and high-speed algorithms.

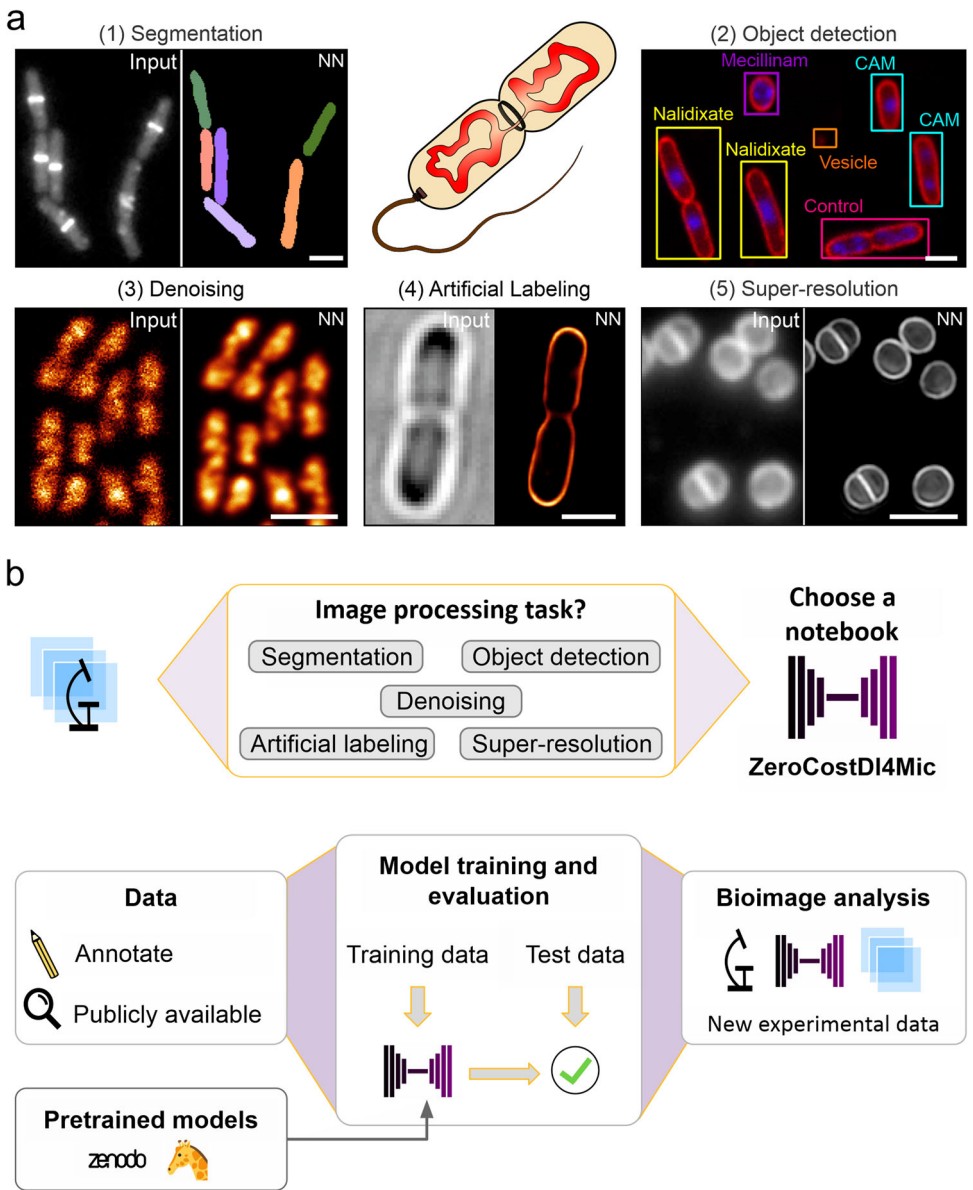

**Fig. 1 Overview of the DL tasks and datasets used in DeepBacs. a** We demonstrate the capabilities of DL in microbiology for segmentation (1), object detection (2), denoising (3), artificial labelling (4) and prediction of super-resolution images (5) of microbial microscopy data. A list of datasets can be found in Supplementary Table 1, comprising different species such as *B. subtilis* (1), *E. coli* (2–4) and *S. aureus* (5) and imaging modalities (widefield (1,2) and confocal (2,3) fluorescence microscopy, bright field imaging (1,2,4) or super-resolution techniques (4,5)). NN: neural network output. CAM = Chloramphenicol. Scale bars: 2 μm. **b** Schematic workflow of applying a DL network. Users select a ZeroCostDL4Mic notebook based on the image analysis task to be performed. Custom annotated or publicly available datasets are used to train and validate DL models. The user can train the DL model from scratch or load a pretrained model from public repositories (e.g., Zenodo or BioImage Model Zoo[77]) and fine tune it. After model accuracy assessment, trained models can be applied to new experimental data.

## Results

In the following sections, we describe the individual datasets (Supplementary Table 1) that we used to perform the tasks shown in Fig. 1. We explain how we designed our experiments, how data was prepared and analysed, and showcase results for the different image analysis tasks. The datasets and selected trained models are publicly available via the Zenodo data sharing platform[38] (Supplementary Table 2), allowing users to explore the DL technology described in this work. An overview of the networks applied in this study is provided in Table 1.

**Image segmentation.** Image segmentation represents the main application of DL technology for bacterial bioimages. Most

networks for bacterial segmentation focus on phase-contrast images as their high contrast allows for efficient segmentation even at high cell densities[33,39,40]. If no phase contrast is available, researchers have to perform segmentation on bright field or fluorescence images, which is challenging due to reduced contrast and increased image heterogeneity. We thus sought to test various DL networks for their capability to segment different types of non-phase-contrast images frequently encountered in microbiological studies. For this, we generated and annotated different datasets comprising bright field and fluorescence microscopy images of rod- and cocci-shaped bacteria (*E. coli* and *S. aureus* for bright field, *S. aureus* and *B. subtilis* for fluorescence) (Fig. 2a). For all datasets, we trained DL models using ZeroCostDL4Mic, as it provides rapid and straight-forward access to a range of popular

**Table 1 Overview of the deep learning models used in this study.**

| Network | Description |
|---|---|
| U-Net | The U-Net, an encoder-decoder type of convolutional neural network (CNN), was proposed for the first time by Olaf Ronneberger to segment microscopy images[41]. It represents a milestone in the field of computer vision (CV), and particularly, for bioimage analysis. The subdivision into two parts, the encoder and the decoder, is the main difference to previous CNN. The encoder extracts features at different scales by successively processing and downscaling the input image. By this process, the content of the input image is projected in more abstract feature space (i.e., feature encoding). Then, these features are upscaled, processed again and synthesised until reaching an image of similar size as the original one but which contains only the information of interest (i.e., feature decoding). In the decoding process, the image features are compared to the respective encoder part image to allow for better adjustment of the output to the initial image. |
| CARE | Content-aware image restoration (CARE) is a supervised DL-based image processing workflow developed by Weigert et al. for image restoration[14]. It uses a U-Net as backbone network architecture and its training parameters are modified to retain intensity differences instead of creating probability masks for segmentation. CARE's main applications are image denoising and resolution enhancement, both in 2D and 3D. CARE is accessible through the CSBDeep toolbox, which allows the deployment of trained models in Fiji as well. |
| StarDist | StarDist was developed by Schmidt et al. for the supervised segmentation of star-convex objects in 2D and 3D (i.e., ellipse-like shaped objects)[9]. StarDist uses a U-Net like CNN to detect the centroid of each object in an image and the distances from each centroid to the object boundary. These features allow the representation of each object boundary as a unique polygon in the image, which is then used to determine the object segmentation. By treating each detected object independently, StarDist achieves an excellent performance at high object densities. The StarDist Python package is optimised for a fast and reliable training and prediction. StarDist is available as Fiji and Napari plugins and it is also integrated in QuPath[85]. The battery of StarDist software is equipped with pretrained models for the segmentation of cell nuclei in fluorescence and histology images. |
| SplineDist | SplineDist was developed by the Uhlmann's group and represents an extension of StarDist to detect non-star-convex objects[42]. The latter is achieved by substituting the polygons by splines, which enables the reconstruction of more complex structures besides ellipses. SplineDist is equipped with a Python package for training and deployment of their models. |
| pix2pix | The supervised pix2pix, developed in the lab of Alexei Efros, belongs to the class of generic adversarial networks (GANs)[18]. Two separate U-Net-like networks are trained in parallel to perform image-to-image translation (e.g., to convert daylight to nightlight photos, from DIC to fluorescence). Hereby, one network performs the image translation task (i.e., generator), while the second network tries to discriminate between the predicted image and the ground truth (i.e., discriminator). Model training is considered successful when the discriminator cannot distinguish between the original and the generated image anymore. In microscopy, pix2pix is employed to generate super-resolved images from sparse PALM data (ANNA-PALM)[86] or to convert low-resolution (widefield, confocal) to high-resolution images (confocal, STED)[16]. |
| Noise2Void | Noise2Void is a self-supervised network for image denoising proposed in microscopy by the Jug lab[15]. The idea behind this approach is that each image has a unique noise pattern. Hence, a small portion of an image is used during the training to determine it and then, use it to denoise the entire image. Training and prediction are fast. Noise2Void is part of the CSBDeep toolbox as well and in this particular case, training and deployment are available in Fiji. |
| YOLOv2 | YOLOv2 was developed by Redmon and Farhadi for supervised (real-time) detection and classification of objects in images[12]. Training requires paired data consisting of images and corresponding bounding boxes (i.e., rectangles drawn around the object of interest). For fast performance, YOLOv2 divides images into grids, in which each segment can only contain the centroid of a single object. |
| fnet | fnet was developed by the group of Gregory Johnson for artificial labelling of bright field/transmitted light images[20]. In the original work, fnet generated pseudo-fluorescent images of different organelles in individual image stacks, increasing the multiplexing capability and reducing phototoxicity. As a U-Net type network, it performs supervised learning and requires the bright field/transmitted light images and fluorescence images as input. Originally designed for 3D images, we deploy a variant that can be used for artificial labelling of 2D images. |

DL networks[22]. To evaluate their performance, we calculated common metrics, which compare the network output of test images to the respective annotated ground truth (Supplementary Note 1, Table 2).

Five popular networks were used for segmentation, namely U-Net[8,41], CARE[14], pix2pix[18], StarDist[9] and its recent variant SplineDist[42] (Supplementary Table 3). As the underlying network architectures vary, the workflows to obtain instance segmentations (individual objects from binary masks) differ in terms of, for example, input/output data formats and image post-processing (Supplementary Fig. 1). While StarDist and SplineDist provide instance segmentation directly as network output, instances have to be generated from U-Net, CARE and pix2pix predictions by post-processing the outputs. CARE and pix2pix were not explicitly designed for segmentation, but are versatile enough to generate probability maps that can be segmented subsequently. Similar to U-Net, the instance segmentation performance depends not only on the trained model but also on the applied post-processing routine.

For our first dataset, we recorded live *S. aureus* cells immobilised on agarose pads, either in bright field mode or using the fluorescent membrane stain Nile Red (Fig. 2b). Due to their coccoid shape, we speculated that StarDist[9] is well suited to segment this kind of data (Table 1). Testing an unseen and fully annotated dataset demonstrated cell counting accuracies (recall) of $100 \pm 1\%$ (membrane fluorescence) and $87 \pm 3\%$ (bright field) (Table 3). The reduced accuracy for bright field images is caused by optical artefacts at high cell density, leading to merging of defocussed cells (Fig. 2ai). Next to performing segmentation in the cloud, trained models can also be downloaded and conveniently used with the StarDist plugin distributed via the image analysis platform Fiji[43] (Supplementary Video 1). Similar to the ZeroCostDL4Mic notebook, this enables efficient segmentation of live cell time-lapse data (Supplementary Video 2).

Motivated by this finding, we sought to know whether StarDist is also suitable for segmenting rod-shaped cells. For this, we recorded bright field time-lapse images of live *E. coli* cells immobilised under agarose pads[44] (Fig. 2c). Bright field images show less contrast than phase contrast images and suffer from high noise, making them challenging to segment. Still, they are widespread in bacterial imaging, and proper segmentation would be beneficial for studying bacterial proliferation (i.e. cell counts over time) and morphology (cell dimensions and shape). We annotated individual image frames spread over the entire time

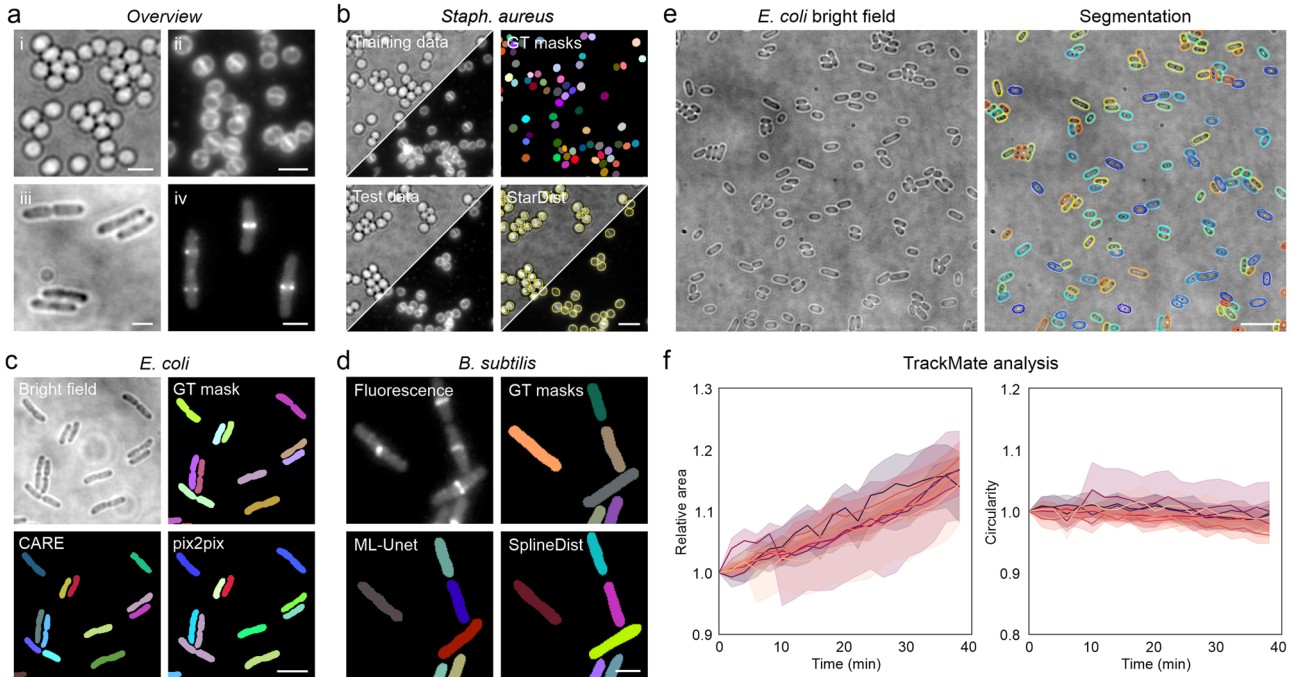

**Fig. 2 Segmentation of bacterial images using open-source deep learning approaches. a** Overview of the datasets used for image segmentation. Shown are representative regions of interest for (i) *S. aureus* bright field and (ii) fluorescence images (Nile Red membrane stain), (iii) *E. coli* bright field images and (iv) fluorescence images of *B. subtilis* expressing FtsZ-GFP[47]. **b** Segmentation of *S. aureus* bright field and membrane-stain fluorescence images using StarDist[9]. Bright field and fluorescence images were acquired in the same measurements and thus share the same annotations. Yellow dashed lines indicate the cell outlines detected in the test images shown. **c** Segmentation of *E. coli* bright field images using the U-Net type network CARE[14] and GAN-type network pix2pix[18]. A representative region of a training image pair (bright field and GT mask) is shown. **d** Segmentation of fluorescence images of *B. subtilis* expressing FtsZ-GFP using U-Net and SplineDist[42]. GT = ground truth. **e** Segmentation and tracking of *E. coli* cells during recovery from stationary phase. Cells were segmented using StarDist and tracked with TrackMate[45,46]. **f** Plots show the mean (line) and standard deviation (shaded areas) for all cells in seven different regions of interest (colour-coded). Morphological features were normalised to the first value for each track. Scale bars are 2 μm (**a**, **d**), 3 μm (**b**, **c**) and 10 μm (**e**).

---

**Table 2 Metrics to evaluate model performance.**

| Metric | Description |
| --- | --- |
| Intersection-over-Union (IoU) | The IoU metric reports on the overlap of output and ground truth segmentation masks. Higher overlap represents a better agreement between the model output and ground truth. |
| Precision and recall | These metrics are used to quantify the performance of instance segmentation or object detection. Precision is a measure for the specificity and describes which fraction of the detected objects are correctly detected/assigned. Recall, on the other hand, describes the sensitivity, i.e. how many objects out of all objects in the dataset were detected. |
| (mean) average precision ((m)AP) | This metric is used to evaluate model performance in object detection and classification tasks. It describes the models' ability to detect objects of individual classes (AP) or all classes (mAP) present in the dataset. To obtain the average precision, precision and recall values for the individual object classes are calculated at different detection thresholds. mAP is calculated by averaging all single-class AP values. |
| Structural similarity (SSIM) | The SSIM value quantifies how similar two images are with respect to pixel intensities and intensity variations. As it is calculated locally using a defined windows size, it provides a similarity map that allows to identify regions of high or low similarity. |
| Peak-signal-to-noise ratio (PSNR) | The PSNR metric compares the signal to noise ratio of images with lower signal-to-noise to the high SNR counterpart based on the pixel-wise mean squared error. It is often used to compare the results of image compression algorithms, but can also be applied to evaluate model performance on paired test data. |

---

series and trained supervised DL networks to reflect varying cell sizes and densities. All networks showed good performance for semantic segmentation, as indicated by high intersection-over-union (IoU, see Supplementary Note 1, Table 3) values for all time points (IoU > 0.75) (Supplementary Fig. 2a). For instance segmentation, however, the model performance varied strongly. While instance segmentations of U-Net, CARE and pix2pix worked well for early time points and thus low cell density (Fig. 2b), individual cells in crowded regions could not be

resolved using basic image post-processing (i.e. thresholding). This led to a successive decrease in the recall value over imaging time and therefore a decreasing number of correctly identified cells (Supplementary Fig. 2a). The best counting performance at low and high cell densities was achieved using StarDist, which correctly identified 87% of the cells for the entire test dataset (Table 3). However, as StarDist assumes star-convex shaped objects, the accuracy of the predicted cell shape decreases with increasing cell length (and thus axial ratio), rendering this

**Table 3 Summarised network performance for the different tasks and datasets.**

| Task | Organism | Dataset | Figure | Network | Network performance | | | | | |
| --- | --- | --- | --- | --- | --- | --- | --- | --- | --- | --- |
| | | | | | IoU | Precision | Recall | mAP | SSIM | PSNR |
| Segmentation | S. aureus | Bright field | 2B | StarDist | 0.64 ± 0.01 | 0.90 ± 0.03 | 0.87 ± 0.03 | - | - | - |
| | S. aureus | fluorescence | 2B | StarDist | 0.91 ± 0.03 | 0.98 ± 0.02 | 1.00 ± 0.01 | - | - | - |
| | E. coli | Bright field | 2C | U-Net | 0.82 ± 0.03 | 0.56 ± 0.20 | 0.39 ± 0.20 | - | - | - |
| | E. coli | Bright field | 2C | ML-U-Net | 0.78 ± 0.05 | 0.71 ± 0.11 | 0.81 ± 0.09 | - | - | - |
| | E. coli | Bright field | 2C | CARE | 0.83 ± 0.03 | **0.85 ± 0.06** | 0.78 ± 0.09 | - | - | - |
| | E. coli | Bright field | 2C | StarDist | 0.78 ± 0.03 | 0.83 ± 0.12 | **0.87 ± 0.07** | - | - | - |
| | E. coli | Bright field | 2C | pix2pix | **0.86 ± 0.02** | 0.82 ± 0.07 | 0.64 ± 0.12 | - | - | - |
| | B. subtilis | fluorescence | 2D | U-Net | 0.78 ± 0.06 | 0.67 ± 0.21 | 0.63 ± 0.26 | - | - | - |
| | B. subtilis | fluorescence | 2D | ML-U-Net | **0.82 ± 0.02** | 0.79 ± 0.16 | 0.82 ± 0.20 | - | - | - |
| | B. subtilis | fluorescence | 2D | CARE | 0.74 ± 0.04 | 0.44 ± 0.28 | 0.36 ± 0.23 | - | - | - |
| | B. subtilis | fluorescence | 2D | StarDist | 0.76 ± 0.03 | **0.88 ± 0.05** | **0.92 ± 0.05** | - | - | - |
| | B. subtilis | fluorescence | 2D | SplineDist | 0.72 ± 0.04 | 0.88 ± 0.06 | 0.87 ± 0.10 | - | - | - |
| | B. subtilis | fluorescence | 2D | pix2pix | 0.69 ± 0.07 | 0.69 ± 0.20 | 0.64 ± 0.21 | - | - | - |
| | all above | mixed model | S2 | StarDist | 0.74 ± 0.06 | 0.88 ± 0.08 | 0.84 ± 0.14 | - | - | - |
| | E. coli | Bright field (stat. Phase) | 2E | StarDist | 0.83 ± 0.02 | 0.95 ± 0.04 | 0.97 ± 0.03 | - | - | - |
| Object detection | E. coli | Growth stage (large FoV) | - | YOLOv2 | - | 0.65 ± 0.10 | 0.47 ± 0.09 | 0.39 ± 0.09 | - | - |
| | E. coli | Growth stage (small FoV) | 3A | YOLOv2 | - | 0.73 ± 0.03 | 0.74 ± 0.08 | 0.67 ± 0.10 | - | - |
| | E. coli | Antibiotic profiling | 3B | YOLOv2 | - | 0.76 ± 0.13 | 0.76 ± 0.23 | 0.66 ± 0.23 | - | - |
| Denoising | E. coli | H-NS-mScarlet-I | 4A | PureDenoise | - | - | - | - | 0.834 ± 0.013 | 33.5 ± 0.9 |
| | E. coli | H-NS-mScarlet-I | 4A | Noise2Void | - | - | - | - | 0.881 ± 0.005 | 34.9 ± 0.9 |
| | E. coli | H-NS-mScarlet-I | 4A | CARE | - | - | - | - | **0.897 ± 0.005** | **36.1 ± 0.9** |
| | E. coli | MreB-sfGFP | 4E | PureDenoise | - | - | - | - | 0.458 ± 0.013 | 26.2 ± 0.9 |
| | E. coli | MreB-sfGFP | 4E | CARE | - | - | - | - | **0.520 ± 0.010** | **27.0 ± 0.8** |
| | B. subtilis | FtsZ | 4G | Noise2Void | - | - | - | - | - | - |
| Artificial labelling | E. coli | Widefield | 5A | CARE | - | - | - | - | 0.83 ± 0.05 | 24.4 ± 1.2 |
| | E. coli | Widefield | 5A | fnet | - | - | - | - | **0.88 ± 0.06** | **25.9 ± 1.7** |
| | E. coli | PAINT | 5A + S8 | CARE | - | - | - | - | **0.85 ± 0.05** | 24.0 ± 1.2 |
| | E. coli | PAINT | 5A + S8 | fnet | - | - | - | - | **0.85 ± 0.07** | **24.3 ± 1.4** |
| Resolution enhancement | E. coli | WF/SIM | 6A | CARE | - | - | - | - | 0.84 ± 0.03 | 25.4 ± 1.0 |
| | S. aureus | WF/SIM | 6B | CARE | - | - | - | - | 0.92 ± 0.01 | 28.2 ± 0.7 |

Bold numbers mark the best-performing network when multiple networks were applied to the same dataset.

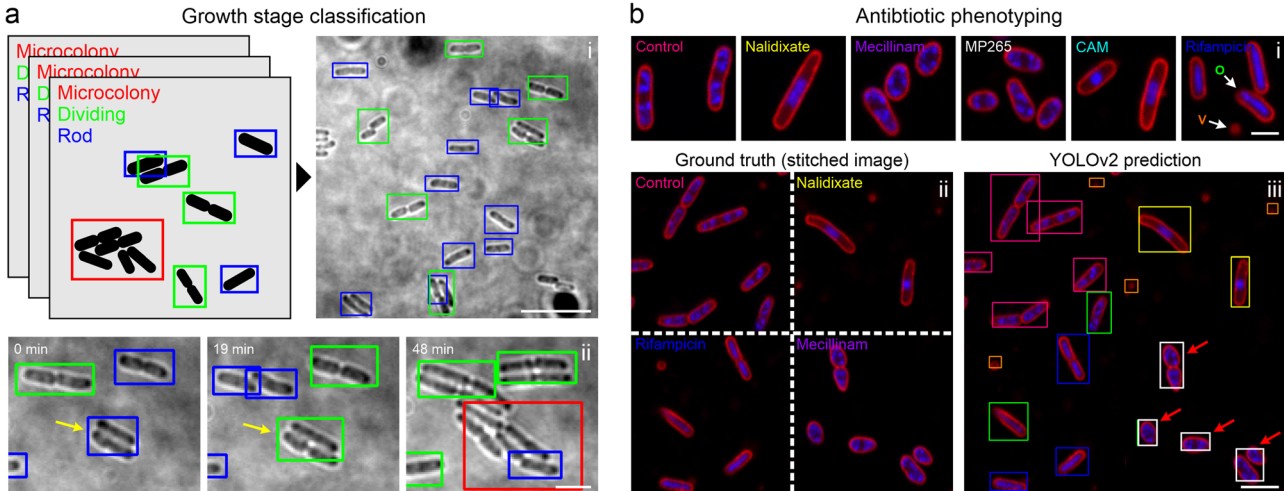

**Fig. 3 DL-based object detection and classification. a** A YOLOv2 model was trained to detect and classify different growth stages of live *E. coli* cells (i). "Dividing" cells (green bounding boxes) show visible septation, the class "Rod" (blue bounding boxes) represents growing cells without visible septation and regions with high cell densities are classified as "Microcolonies" (red bounding boxes). (ii) Three individual frames of a live cell measurement. **b** Antibiotic phenotyping using object detection. A YOLOv2 model was trained on drug-treated cells (i). The model was tested on synthetic images randomly stitched from patches of different drug treatments (ii). Bounding box colours in the prediction (iii) refer to the colour-code in (i). Vesicles (V, orange boxes) and oblique cells (O, green boxes) were added as additional classes during training. Mecillinam-treated cells were misclassified as MP265-treated cells (red arrows). Scale bars are 10 μm (**a**, overview), 3 μm (lower panel in **a** and **b**) and 1 μm (**b**, upper panel).

network less suited for morphometry of elongated rod-shaped cells (Supplementary Fig. 2b). Using a multi-label U-Net (ML-U-Net, trained to detect cell cytosol and boundary) instead of a single-label U-Net provided the best compromise between instance segmentation performance and proper prediction of cell morphology. This notebook uses training data, in which each label (here cell boundary and cytosol) is visualised by a different grey value. Such training data can be generated using Fiji macros that we provide in our repository, and which also allow to obtain instance segmentation from the network output. Applying the trained ML-U-Net to time-lapse videos allows to extract single-cell instances that can subsequently be tracked using e.g. TrackMate[45,46] (Supplementary Video 3).

Next, we were interested in the performance of DL networks for the segmentation of complex fluorescence data. Although typical images used for segmentation show high contrast (fluorescent membrane stains, phase-contrast images), images with complex fluorescence distributions or low signal represent a significant challenge. Fluorescence images of *B. subtilis* cells expressing FtsZ-GFP[47] show a bimodal intensity distribution with the characteristic localisation in the septal region and diffusing FtsZ monomers that produce dim labelling of the cytosol (Fig. 2d). Growth for several cell cycles results in the multi-cellular chains of *B. subtilis* and microcolony formation, providing a dataset with increasing cell density and a large number of cell-to-cell contacts. When we tested different networks for this challenging dataset, we found that U-Net and pix2pix provided well segmentable predictions at low to medium cell density (Fig. 2d, Supplementary Fig. 2c, d). However, these networks also suffered from undesired cell merging at high cell densities, leading to reduced recall and precision values (Supplementary Fig. 2c, d). As for segmentation of *E. coli* bright field images, StarDist and its variant SplineDist[42] showed high recall and precision values also for mid- and high-density regions, while the multi-label U-Net preserved cell morphology at slightly lower instance segmentation accuracy (Supplementary Fig. 2d). In contrast to StarDist, SplineDist is not limited to convex shapes, which makes it a good candidate network for the segmentation of curved bacteria (e.g., *Caulobacter crescentus*). Nevertheless,

SplineDist is computationally more expensive, and thus takes longer to train.

As another example for cell tracking, we trained a StarDist model to detect stationary *E. coli* cells (bright field images) that resume growth upon addition to agarose pads (Fig. 2e). Applying this model to time-lapse data and tracking individual cells over time (Supplementary Video 4) allows to extract morphological features such as cell area or circularity. Our analysis revealed that the cell area (and thus also the volume) increases with growth, while the circularity only decreases slightly (Fig. 2f). This might indicate that bacteria expand in all directions during lag phase, in contrast to exponentially growing cells that mainly elongate (Supplementary Video 3). Of note, StarDist models can be directly applied within the Fiji TrackMate plugin, making cell tracking straight-forward and convenient[46].

Finally, and motivated by generalist approaches such as Cellpose[10], we were interested in whether a single StarDist model is capable of performing all the segmentation tasks shown in Fig. 2b-d. We thus trained a model on pooled training data and evaluated its performance (Supplementary Fig. 3). Although bright field and fluorescence images differ significantly, the obtained 'all-in-one' model showed similar precision and recall values compared to the specialist models (Supplementary Table 4). However, it also shows the same limitations, such as incomplete predictions for long and curved bacteria (Supplementary Fig. 3). For cells with suitable morphology, StarDist allows segmenting large images with thousands of cells, as shown for live, rod-shaped *Agrobacterium tumefaciens* cells imaged at various magnifications (Supplementary Fig. 4). It also demonstrates that DL segmentation models can perform well on images with low signal and a noisy background.

**Object detection and classification.** To explore the potential of object detection for microbiological applications, we employed an implementation of YOLOv2[12] for two distinct tasks: Identification of cell cycle events such as cell division in bright field images (Fig. 3a) and antibiotic phenotyping of bacterial cells based on membrane and DNA stains (Nile Red and DAPI, respectively)

(Fig. 3b) (Supplementary Table 5). These labels are commonly used to study antibiotic action, as they are easy to use and also facilitate live-cell staining of bacterial cells[48].

We chose YOLOv2 due to its good performance in a recent study, in which a network was trained to classify cell nuclei in fluorescence images[34]. For growth stage classification of live *E. coli* cells in bright field images we used the same dataset employed for segmentation (Fig. 2c). Here, we wanted to discriminate between rod-shaped cells, dividing cells, and microcolonies (defined as 4+ cells in close contact) (Fig. 3a i) using a training and test dataset that we annotated online (https://www.makesense.ai/) or locally (LabelImg)[49] (see methods).

Due to the small size of bacterial cells, we initially investigated the influence of the object size on the performance of YOLOv2. Since object detection networks rescale input images to a defined size, the relative object size changes with size of the region of interest. When we trained our model on large images, we encountered missed objects, wrong bounding box positioning or false classifications (Supplementary Video 5). To quantify this effect, we determined recall and precision values as well as mean average precision (mAP, see Supplementary Note 1 and Table 2) for the test dataset (Table 1, Supplementary Table 6). mAP represents the common metric for object detection, taking into account model precision and recall over a range of object detection thresholds[12]. For object detection challenges (e.g. PASCAL visual object classes (VOC) challenge[50], well-performing models typically yield mAP values in the range of 0.6–0.8). However, for our large-FoV growth stage classification dataset, we obtained a mAP of 0.386, with a size-dependent performance for the different classes ($AP_{Microcolony} > AP_{dividing} > AP_{rod}$ (non-dividing)) (Supplementary Table 6). Smaller images resulted in improved network performance (mAP = 0.667) with classification of the majority of cells in the image (Fig. 3a, Supplementary Video 6). Knowing the size-dependent performance is important for the design of an object detection experiment. If the focus is on the detection of large structures, such as microcolonies, the YOLOv2 model can be trained on large fields of view. However, small regions of interest or higher optical magnification should be used if small objects are to be detected. Object density is another parameter that affects the performance of YOLOv2 models. As YOLOv2 uses a grid-based approach for object detection, in which each grid region can only hold one object, very close objects (i.e. non-dividing cells at $t = 0$ in Fig. 3a or dividing cells at $t = 19$ min, yellow arrows) are not resolved. Instead, only one bounding box of the corresponding class is predicted. Thus, object density should be considered as a limiting factor when planning to train a network for object detection. When applied to time-lapse recordings of growing *E. coli* cells, the model facilitates identification of class transitions, e.g. from rod-shaped (non-dividing) cells to dividing cells and at later time points to microcolonies (Fig. 3a ii, Supplementary Video 6).

As a second task for object detection, we explored its suitability for antibiotic phenotyping (Fig. 3b). In antibiotic phenotyping, bacterial cells are classified as non- or drug-treated cells based on cell morphology and subcellular features (commonly DNA and membrane stains). This facilitates the assignment of a mode of action to antibiotics or potential candidate compounds, being a promising tool in drug discovery[3,48]. To explore whether object detection networks can be used for this purpose, we generated a dataset of images including membrane- and DNA-labelled *E. coli* cells grown in the absence or presence of antibiotics. We used five different antibiotics that target different cellular pathways (Fig. 3b, Supplementary Table 7). Nalidixate blocks DNA gyrase and topoisomerase IV, thus stalling DNA replication. Mecillinam and MP265 (a structural analogue of A22) perturb cell morphology by

inhibiting peptidoglycan crosslinking by PBP2 or MreB polymerisation, while rifampicin and chloramphenicol inhibit transcription and translation, respectively. As additional classes, we included untreated cells (control), membrane vesicles and oblique cells. The latter class represents cells that are only partially attached to the surface during immobilisation. Such cells can be identified by a focus shift and are present in all growth conditions (Fig. 3b i). Further examples for each class are provided in Supplementary Fig. 5.

We trained a YOLOv2 model on our annotated dataset and tested its performance on images containing cells treated with different antibiotics (stitched images, see methods) (Fig. 3b ii/iii) or images that only show one condition similar to the training dataset (Supplementary Fig. 6). The YOLOv2 model showed a comparable performance for both datasets with mAP values of 0.66 (stitched image dataset) and 0.69 (individual conditions), indicating a good generalisability of our model. Hereby, the AP values for the different classes varied substantially, ranging from 0.21 (vesicles) to 0.94 (control) (Supplementary Table 8). Poor prediction of membrane vesicles is likely caused by their small size, which agrees with the observations made for growth stage prediction. Intermediate AP values are observed when antibiotics induce similar morphological changes, as it is the case for mecillinam (AP = 0.605) and MP265 (AP = 0.526). This led to misclassification between these classes (Fig. 3b, red arrows, Supplementary Fig. 6), indicating that both treatments result in a highly similar phenotype.

This similarity allowed us to test whether YOLOv2 can identify antibiotic modes of actions in unseen images. We omitted MP265 data during model training, but included images of MP265-treated cells in the test data. Due to their similar phenotype MP265-treated cells should hence be predicted as Mecillinam-treated. This was indeed the case, as shown by the high mAP value (0.866) and specificity (recall = 0.961) (Supplementary Fig. 6b), demonstrating the applicability of object detection networks for mode-of-action-based drug screening.

**Denoising**. As denoising approaches allow for faster and more gentle imaging[14,15,35,51], we consider them as powerful tools for bacteriology. To test their applicability to bacterial data, we recorded paired low and high signal-to-noise ratio (SNR) images of an H-NS-mScarlet-I[52] fusion protein in live *E. coli* cells. H-NS decorates the bacterial nucleoid homogeneously under nutrient-rich growth conditions and maintains nucleoid association after chemical fixation[53]. This allows the study of chromosome organisation and dynamics, an important field of bacterial cell biology. We trained the CARE and N2V models on image pairs acquired using chemically fixed cells to prevent motion blur in the training dataset (Supplementary Table 9). We found that both parametric and DL-based approaches strongly increased the SNR of noise-corrupted images, as indicated by the peak signal-to-noise ratio (PSNR) and structural similarity index (SSIM)[54] (Fig. 4a). These metrics are commonly used to assess SNR and quality of image pairs, with higher values representing improved performance (Supplementary Note 1, Table 2). Under the conditions tested, we obtained the best results using the supervised network CARE (SSIM = $0.897 \pm 0.005$, PSNR = $36.1 \pm 0.09$) (Table 3). Next, we applied the trained models to denoise live-cell time series recorded under low-SNR conditions (Fig. 4b, Supplementary Video 7). This led to an apparent increase in SNR, and intensity analysis revealed a 20x lower photobleaching rate as indicated by the exponential intensity decay time $t_{1/2}$ (Fig. 4c).

However, the performance of the different denoising approaches in fast live cell measurements could not be assessed by the standard PSNR and SSIM metrics because of the lack of

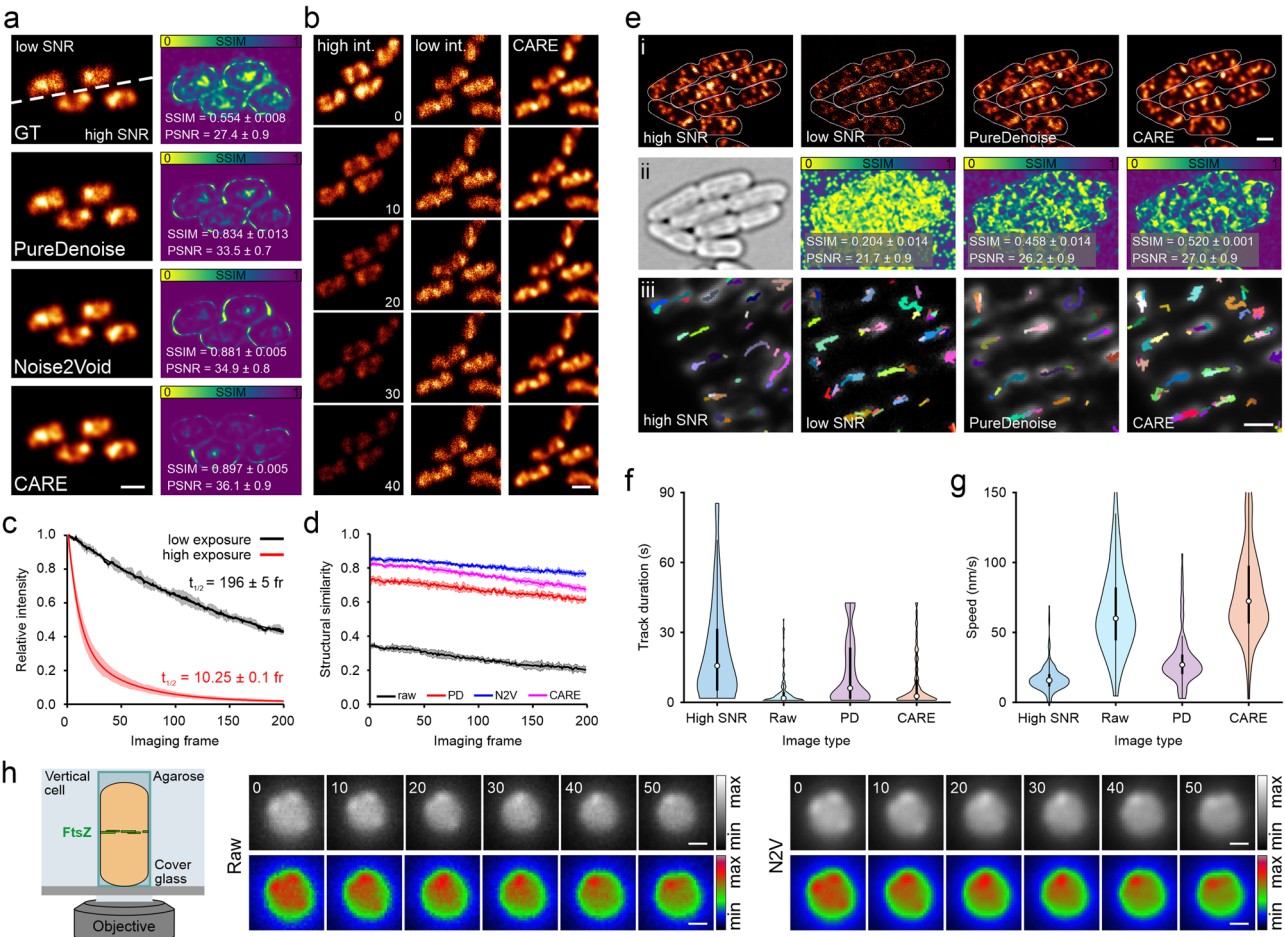

**Fig. 4 Image denoising for improved live-cell imaging in bacteriology. a** Low and high signal-to-noise ratio (SNR) image pairs (ground truth, GT) of fixed *E. coli* cells, labelled for H-NS-mScarlet-I. Denoising was performed with PureDenoise (parametric approach), Noise2Void (self-supervised DL) and CARE (supervised DL). Structural similarity (SSIM) maps compare low-SNR or predictions to ground truth (GT) high-SNR data. **b** 10 s interval representative time points of a live-cell measurement recorded at 1 Hz frame rate, demonstrating CARE can provide prolonged imaging at high SNR using low-intensity images as input. $t_{1/2}$ represents the decay half time. **c** Intensity over time for different imaging conditions providing low/high SNR images shown in **a**/**b**. **d** Structural similarity between subsequent imaging frames was calculated for raw and restored time-lapse measurements (Methods). **e** Denoising of confocal images of MreB-sfGFP[sw] expressing *E. coli* cells, imaged at the bottom plane (i). Outlines show cell boundaries obtained in transmitted light images (ii). (ii). Transmitted light image and SSIM maps generated by comparison of raw or denoised data with the high SNR image. (iii) Tracks of MreB filaments (colour-coded) and overlaid with the average image (grey) of a live-cell time series. Violin plots show the distribution of track duration (**f**) and speed (**g**) for the high SNR, low SNR (raw) and denoised image series, with mean values denoted by circles and percentiles by black boxes. Note that the distribution in **g** was cut at a max speed of 150 nm/s, excluding a small number of high-speed outliers but allowing for better visualisation of the main distribution. **h** Denoising of FtsZ-GFP dynamics in live *B. subtilis*. Cells were vertically trapped and imaged using the VerCINI method[47]. Details are restored by Noise2Void (N2V), rainbow colour-coded images were added for better visualisation. Values in **a** and **e** represent mean values derived from 2 (a) and 5 (e) images and the respective standard deviation. **c**, **d** Show mean values and respective standard deviations from 3 measurements. **f**, **g** Show tracking results from individual time series. Scale bars are 1 µm (**a**, **b**, **e** i) and 0.5 µm (**e** iii and **h**).

paired high-SNR images. As noise reduces contrast and structural information content in images, subsequent image frames in low-SRN time series should exhibit higher signal variation than in their high-SNR counterparts. We therefore speculated that calculating the structural similarity between successive image frames (e.g. between frame 1 and frame 2, frame 2 and frame 3, etc.) could report on denoising performance for live-cell time series (Fig. 4d). In fact, all denoising approaches significantly increased SSIM values while preserving relative intensities over time (Supplementary Fig. 7a).

As all the previous models were trained on fixed-cell data, the results on live-cell data could be compromised by potential fixation artefacts. Because N2V is self-supervised, it was possible to train it directly on the live-cell data. Hence, we could compare the performance of a N2V model trained on fixed-cell images

with the one trained on live-cell images. This resulted in high structural similarity throughout the time series (Supplementary Fig. 7b), indicating that no artefacts were introduced by training on fixed-cell data. Similar observations were made when comparing the fixed-cell N2V and CARE models. Analysis of raw and denoised (CARE) time series of chemically fixed cells showed a constant SSIM value of 0.96 in the subsequent-frame analysis (Supplementary Fig. 7c). This indicates that the high contribution of shot-noise under low-SNR conditions for this target can be overcome by the denoising method. Of note, the SSIM value obtained in fixed-cell measurements is higher than for denoised live-cell time series (0.82) (compare Fig. 4d and Supplementary Fig. 7c). To test whether this effect is caused by nucleoid dynamics (Supplementary Video 7), we recorded a time series under high-SNR imaging conditions using a small region of

interest. High SNR imaging leads to lower noise contribution and higher SSIM values for subsequent image frames, but induces strong photobleaching that leads to a rapid drop in structural similarity over time (Fig. 4b, Supplementary Fig. 7d). However, the first SSIM value of the high-SNR time series (representing the similarity between frame 1 and frame 2) is close to the corresponding SSIM value of the denoised low-SNR time series. This indicates that (i) the model provides optimal denoising performance and (ii) the lower SSIM values in live-cell measurements originate from nucleoid dynamics rather than representing denoising artefacts.

To probe the limits of denoising approaches, we recorded confocal images of *E. coli* cells chromosomally expressing MreB-sfGFP[sw][55]. MreB forms filaments, which show processive movement along the inner membrane (Fig. 4e)[56–58]. Imaging the bottom plane of the cells, this movement can be tracked in order to determine filament speed, an analysis that is commonly performed in MreB studies. We speculated that processive movement represents a good readout for model performance, and trained a CARE model on paired low-SNR/high-SNR data. The trained model, as well as the parametric denoising using PureDenoise, improved the SNR and structural similarity, although to a smaller extent compared to the HNS-mScarlet-I nucleoid dataset (Fig. 4a, Table 3). To test model performance on live-cell time series, we tracked MreB filaments using TrackMate[45,46]. High SNR movies show long tracks and a filament speed of 17 nm/s, which is in good agreement with reported values[59], while tracking raw data resulted in significantly shorter tracks and higher filament speed (Fig. 4f/g). Surprisingly, DL-based denoising did not restore track length and speed, which we account to model hallucinations that are caused by the high contribution of shot noise at this low SNR (Fig. 4f/g). This leads to strong frame-to-frame intensity fluctuations which obscure processive filament movement (Supplementary Video 8). As PureDenoise can integrate temporal information during the denoising process, shot-noise contribution is reduced, leading to more sensible restoration results, i.e. processive filament motion similar to the high SNR measurements (Fig. 4e–g, Supplementary Video 8).

As another example, we denoised time-lapse images of FtsZ treadmilling in live *B. subtilis* cells. These movies were recorded in vertically aligned cells using the so-called VerCINI approach (Fig. 4h) and contributed to study the critical role of FtsZ treadmilling in cell division[47,60]. As the constant movement of FtsZ-GFP renders acquisition of low and high SNR image pairs difficult, we used the self-supervised N2V method for the denoising task. Here, denoising emphasises subtle details that are difficult to be identified in the raw image data (Fig. 4h). This allows for long time-lapse imaging of FtsZ dynamics with enhanced image quality (Supplementary Video 9).

**Artificial labelling**. Artificial labelling can be very useful for bacterial imaging, as it increases the multiplexing potential and circumvents phototoxicity. Because of their much smaller size, bacteria provide less structural information in bright field images than eukaryotic cells (Fig. 5a). However, we regard the cell envelope as a promising target for artificial labelling, as exact cell shape determination is of interest for morphological studies (e.g. for antibiotic treatments) or in the context of spatial positioning of target molecules in individual cells[61,62]. This is even more valuable if super-resolution information is obtained. To explore whether such information can be extracted using DL, we recorded different training datasets. The first dataset includes bright field and corresponding diffraction-limited widefield fluorescence images, in which the *E. coli* membrane is stained by the lipophilic

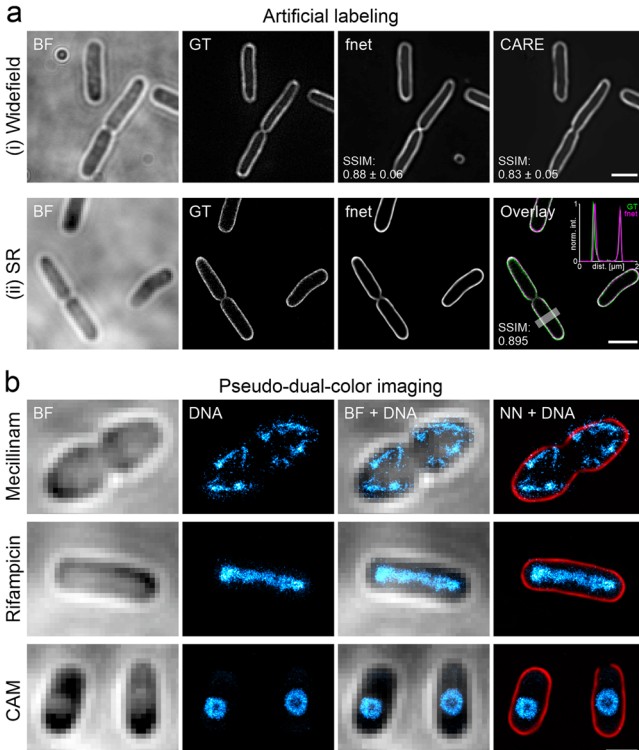

**Fig. 5 Artificial labelling of *E. coli* membranes. a** fnet and CARE predictions of diffraction-limited (i) and PAINT super-resolution (SR) (ii) membrane labels obtained from bright field (BF) images. GT = ground truth. Values represent averages from five test images and the respective standard deviation **b** Pseudo-dual-colour images of drug-treated *E. coli* cells. Nucleoids were super-resolved using PAINT imaging with JF[646]-Hoechst[64]. Membranes were predicted using the trained fnet model. CAM = Chloramphenicol. Scale bars are 2 μm (**a**) and 1 μm (**b**).

dye Nile Red (Fig. 5a, i). For the second dataset, we acquired super-resolved PAINT images[63,64] together with bright field images (Fig. 5a ii). For both datasets, we tested a 2D version of fnet[20], as well as CARE. For the diffraction-limited dataset, both networks were able to predict pseudo-fluorescence images from bright field images, with fnet showing slightly better performance ($SSIM_{fnet} = 0.88 \pm 0.06$, $SSIM_{CARE} = 0.83 \pm 0.05$) (Fig. 5a, Table 3). This is not surprising as fnet was designed for artificial labelling, while the good performance of CARE demonstrates the versatility of this network. Similar values were obtained for the super-resolution dataset (Fig. 5a, Supplementary Fig. 8), with predictions showing good agreement also on the sub-diffraction level (see cross-section as inset in Fig. 5a). Additionally, although trained on fixed cells, the model can also be used to predict highly resolved membrane signal in live-cell time series (Supplementary Video 10).

We then wanted to know how well our model generalises, i.e. whether it can predict the super-resolved membrane of bacteria grown in the presence of different antibiotics (see methods). Both fnet and CARE models successfully predicted the membrane stains in drug-treated cells ($SSIM_{fnet} = 0.85 \pm 0.07$, $SSIM_{CARE} = 0.85 \pm 0.05$, averaged over all treatments) (Table 1), indicating that it detects image features independent from the cell shape (Supplementary Fig. 8). The increased resolution and contrast in membrane predictions allow to map the positioning of subcellular structures (here, the nucleoid) with higher precision (Fig. 5b). This typically requires the acquisition of multi-colour super-resolution images[64], which is more intricate and time-consuming.

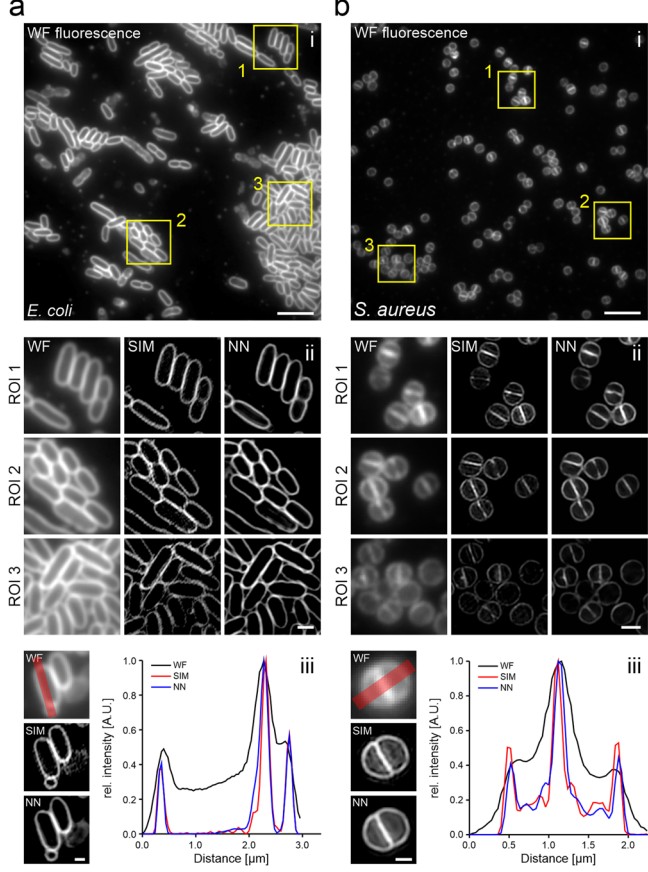

**Fig. 6 Prediction of SIM images from widefield fluorescence images.** Widefield-to-SIM image transformation was performed with CARE for **a** live *E. coli* (FM5-95) and **b** *S. aureus* (Nile Red) cells. Shown are diffraction-limited widefield images (i) and the magnified regions (ii) indicated by yellow rectangles in (i). WF = widefield; NN = neural network output. (iii) Line profiles correspond to the red lines in the WF images and show a good agreement between prediction and ground truth (bottom panel). Scale bars are 10 μm (i), 1 μm (ii) and 0.5 μm (iii).

**Resolution enhancement**. Super-resolution membrane images can also be obtained by training a supervised DL network on paired low-resolution/high-resolution image datasets[14,16,17,22]. Here, we used structured illumination microscopy (SIM)[65] to record membrane images of dye-labelled live *E. coli* and *S. aureus* cells. SIM images are reconstructed from a set of images recorded at different grid positions and angles, which hence requires higher light doses than a single widefield image. As acquisition of such image sets is only required during the network training, but not during its application, super-resolution prediction reduces the light dose and also increases the achievable temporal resolution[17]. Training of two CARE models on paired low/high-resolution images of *E. coli* and *S. aureus* using the ZeroCostDL4Mic notebook provided models that facilitate robust prediction of SIM images from single widefield snapshots (Fig. 6). Here, contrast and resolution of predictions agreed well with reconstructed SIM images (Supplementary Video 11), as shown for cross sections along single *E. coli* and *S. aureus* cells (Fig. 6a, b iii). To evaluate the quality of super-resolution images, we used SQUIRREL, which detects reconstruction artefacts in super-resolution images[66]. This analysis yielded similar errors for neural network predictions compared to SIM reconstructions, both for *E. coli* (resolution-scaled Pearson coefficient of 0.898 ± 0.018 (SIM) vs 0.907 ± 0.018 (prediction)) and *S. aureus* (0.957 ± 0.012 (SIM) vs 0.963 ± 0.010 (prediction))

(Supplementary Fig. 9). SSIM values between predictions and GT SIM images were determined as 0.84 ± 0.03 (*E. coli*) and 0.92 ± 0.01 (*S. aureus*). Estimating the spatial resolution using image decorrelation[67] verified the very good agreement between the predicted (137 ± 7 nm for *E. coli* and 134 ± 5 nm for *S. aureus* images) and reconstructed SIM images (122 ± 2 nm for *E. coli* and 134 ± 1 nm for *S. aureus* images) with the expected 2x increase in resolution (308 ± 24 nm for *E. coli* and 289 ± 5 nm for *S. aureus* widefield images). This strategy is hence well suited to perform single-image super-resolution microscopy[17] in bacterial cells.

## Discussion

In this work we demonstrate the potential of open-source DL technology for the analysis of bacterial bioimages. We employ popular DL networks that were developed by the open-source research community and are implemented in, but not limited to, the user-friendly ZeroCostDL4Mic platform[22]. We used the notebooks as they are provided by the platform, except for the ML-U-Net notebook, which was added to the ZeroCostDL4Mic repository in the frame of this work. This enabled us to perform a variety of different image analysis tasks, such as image segmentation, object detection, image denoising, artificial labelling and the prediction of super-resolution images (Fig. 1). The performance of the networks for each task is provided in Table 3, while a summary on the employed networks and their advantages/disadvantages for bacterial bioimage analysis can be found in Table 4.

Using the datasets that we provide, well-performing models can be trained within the time course of hours (see Supplementary Tables 3, 5, 9-11). Depending on the network, several tens of input images were sufficient, showing that valuable models can be generated even with a limited dataset size and thus moderate effort in data curation. As multiple DL networks exist for each specific tasks (in particular for cell segmentation), and the performance of these networks can vary strongly depending on the images to be analysed[68], there is a need for user-friendly implementations that enable researchers to test the different networks and identify the best-performing network for a particular dataset.

Testing different segmentation networks, we found that Star-Dist and SplineDist are well suited to segment small rod-shaped and coccoid bacteria in bright field and fluorescence images (Fig. 2, Supplementary Figs. 3 and 4), while U-Net and pix2pix performed better for elongated cells at low to mid cell density. The performance at higher densities could be improved by predicting cell boundaries and cytosol using a multi-label U-Net, followed by post-processing of the network output. Integrated into the ZeroCostDL4Mic environment, this notebook allows streamlined training and evaluation of models that can not only provide better segmentation results, but could also be used to discriminate between different object classes.

Having a closer look at the input data can already give indications about which network might be more or less suited for the segmentation task (Table 4, Supplementary Note 2). In our experience, the networks explored in this work are well suited to segment images recorded under standard conditions (e.g. exponential growth phase, regularly shaped cells, narrow size distribution). However, they might be of limited use or require large training datasets for more specialised cases, e.g. studying filamentation, irregularly shaped cells or biofilms. In such cases, we refer to DL networks developed for the particular segmentation task[28–32].

Instance segmentations can subsequently be used for downstream applications such as tracking cell lineages or morphological changes. If this is not already included in the network[32,33], segmentation masks can be used with TrackMate, which was

**Table 4 Advantages and disadvantages of specific approaches for the performed image analysis tasks.**

| Task | Network | Advantages | Disadvantages | Recommended for | Training speed |
|---|---|---|---|---|---|
| (Instance) Segmentation | Classical U-Net | Better feature synthesis and correspondence with the input image when compared with classical fully connected neural networks. Reproducible inference in Fiji. | Requires annotated masks and postprocessing of the network output. | Low cell densities, high contrast, arbitrary cell shapes | Intermediate |
| | Multilabel U-Net | Semantic segmentation (background, cell boundary and cell cytosol) which improves to distinguish touching objects. Reproducible inference in Fiji. | Requires annotated masks and postprocessing of the network output. Implemented for 2D data. | Arbitrary cell shapes | Intermediate |
| | StarDist | Highly generalisable and excellent performance at high object density; available for 2D and 3D; equipped for processing of large field of views; reproducible inference in Fiji, QuPath and Napari. | Limited to star-convex objects, does not work well for objects with large axial ratio (e.g., long rod-shaped cells). | Cocci, Ovococci, small rod-shaped bacterial cells (slow growth, stationary phase), all object densities | Fast |
| | SplineDist | Regularly shaped, non-convex objects | Computationally expensive with a high demand of RAM memory; only implemented for 2D data. | Curved (non-star-convex) objects | Slow |
| | Pix2pix | GAN-type architecture allows for arbitrary image-to-image translation tasks. | Longer training times, post-processing required; high demand of computational resources, risk of strong hallucinations; 2D. | Complex images with multimodal intensity distributions | Slow |
| Object detection | YOLOv2 | Fast training | Limited number of objects per image; low performance for small objects; fails determining objects in highly packed clusters; only available in 2D. | <50 uniformly distributed objects/image | Fast |
| Denoising | CARE | Fast training for 2D and 3D data; the trained model can be deployed in Fiji. | Requires paired data (supervised network). | Targets that allow recording of low/high SNR data (slow or chemically fixed) | Fast |
| | Noise2Void | Unsupervised; new data is used both during the training and inference. Fast training; training and inference available in Fiji. | Lower performance than supervised learning approaches; only available for 2D. | Absence of high SNR images (fast dynamics, labels with low photostability) | (Very) fast |
| | PureDenoise (parametric) | Multi-frame denoising; Fiji plugin; no special requirements and no training required. | Often lower performance than DL-based approaches. | Low SNR data with temporal correlation (e.g., processive movement) | N.A. |

**Table 4 (continued)**

| Task | Network | Advantages | Disadvantages | Recommended for | Training speed |
|---|---|---|---|---|---|
| Artificial labelling | CARE | see above | Lower performance than fnet. | Prediction of membrane labels or structures visible in bright field images | Intermediate |
| | fnet | Training schedule and DL workflow is designed for artificial labelling | - | Prediction of membrane labels or structures visible in bright field images | Intermediate |
| Super-resolution prediction | CARE | see above | Might not predict rare sub-diffraction features | Regular structures (e.g., cell membranes) | Intermediate |

Training speed is only given as a qualitative measure and is based on our experience made during this work. Note that the training time consumption depends on the computational resources available and the size of the training data.

recently updated for the use of DL technology[46] (Supplementary Video 3). The convenient use of StarDist and Cellpose segmentation models directly within TrackMate allows for integrative image analysis of time-lapse data. In this work we tracked *E. coli* cells during release from stationary phase, showing that cells simultaneously expand along the long and short axis (Fig. 2e/f, Supplementary Video 4). For very dense and exponentially growing cells, however, the use DL-based tracking approaches should be favoured, as they strongly reduce erroneous linking of neighbouring cells upon cell division[32,33].

For object detection, we successfully trained models to detect and discriminate cells in specific growth stages (Fig. 3a) or treated with different antibiotics (Fig. 3b). To obtain good YOLOv2 models, the field of view size has to be chosen so that the relative object size matches the networks' receptive field. Analysis of larger images could be obtained by tiling strategies or by employing other networks[13]. Next to their use in post-acquisition image analysis, trained models can also be integrated into smart imaging pipelines in which the microscope system autonomously decides when and/or how to image a particular region of interest. Triggers can be the presence (or dominance) of a specific class[34] or the occurrence of class transitions (for example, initiation of cell division). We anticipate this to be particularly powerful for studying rare events, as smart acquisition strongly reduces data waste and data curation time[34,69]. At the same time, AI-based antibiotic profiling holds great promise for drug-screening applications and antibiotic mode-of-action studies. Although trained on a limited dataset, the YOLOv2 model was able to discriminate between different antibiotic treatments based on its phenological fingerprint (Fig. 3b). We demonstrated that it could already be used for drug screening applications, as it was able to predict a similar mode of action (rod-to-sphere transition) for MP265-treated cells when trained on Mecillinam-treated cells (Supplementary Fig. 6b). As training an object detection network only requires drawing of bounding boxes and no intensive feature design, it can be used straightforwardly by researchers new to this field, especially as membrane and DNA stains are widespread and easy to use. However, we think that the predictive power can be further improved by adding more fluorescence channels, such as indicator proteins that for example report on membrane integrity or the energetic state of the cell. This will result in comprehensive models that can be employed for automatic screening of large compound libraries[3] and might contribute to the discovery of novel antimicrobial compounds, which is desperately needed to tackle the emerging antibiotic resistance crisis[70].

As denoising represents a universally applicable strategy[51], it can be used for any type of sample or microscopy technique. Here, supervised DL networks are the preferred choice (Fig. 4), if well-registered image pairs can be acquired. This is mostly the case for static or slow-moving targets, but acquisition of training data on fixed specimens represents a good alternative[14]. However, we note that this requires proper controls to exclude fixation artefacts (as we have done in previous work[64,71] (Supplementary Fig. 7c)), as these could be learned by the model and erroneously introduced into live-cell data during prediction. For our H-NS-mScarlet-I dataset, we observed a strong increase in image quality and SNR (Fig. 4a). CARE (supervised DL) expectedly outperformed N2V (self-supervised) and PureDenoise (parametric) on our test dataset. Using the trained CARE model on labelled *E. coli* nucleoids in fast-growing cells revealed both high nucleoid complexity and dynamics on the second time-scale (Supplementary Video 6). We observed high-density regions which dynamically move within the area populated by the nucleoid. Such 'super-domains' were reported in previous studies[72,73], eventually representing macrodomains or regions of orchestrated gene expression.

At very low SNR, denoising approaches have to be used with caution and require proper controls. Even though we observed a significant increase in SNR for single MreB images (Fig. 4e), time-lapse videos of processive MreB filament movement revealed strong hallucinations (Supplementary Video 8). These hallucinations arise from structural reconstruction of shot noise and led to artefactual particle tracking results (Fig. 4f/g). The superior performance of PureDenoise (multi-frame denoising) shows that at present parametric approaches can outperform DL-based approaches on data with temporal correlation. However, we anticipate that DL-based multi-frame denoising strategies, as they exist for video denoising[74], will be adapted for bioimage analysis in the near future.

When the acquisition of high-quality data is challenging and no paired high-SNR images are available, self-supervised networks such as Noise2Void can be employed[51]. We show this for time-series of FtsZ-GFP in vertically aligned *B. subtilis* cells[47], in which the gain in SNR allows following subtle FtsZ structures during their treadmilling along the cell septum (Fig. 4h, Supplementary Video 9). Thus, even without access to high-SNR ground truth data, denoising can substantially increase image quality in challenging live-cell data. Concluding, we see the largest benefit of denoising in long-term microscopy experiments and capturing of fast dynamics. These experiments are strongly limited by phototoxic effects, photobleaching, and in temporal resolution, parameters that are improved by denoising approaches. We recommend performing suitable controls and avoiding application to data with excessively low SNR.

We further showed that artificial labelling and prediction of super-resolution images strongly increase the information content of bacterial bioimages. The introduced specificity and improved spatial resolution are particularly useful to study bacterial cells, in which most processes occur on scales close to or below the diffraction limit of light. Training a CARE model on paired bright field and super-resolution membrane images allowed us to artificially label membranes with subpixel accuracy (Fig. 5a). This enables determination of cell size and shapes (morphological analysis) with higher precision, which can be important to describe and compare (deletion) mutants, drug-treated cells or cells grown under different environmental conditions[75]. Correlation of labelled structures to the artificial membrane stain further allows to study intracellular target localisation with high accuracy compared to a bright field image overlay (Fig. 5b). Additionally, as artificial labelling does not require a fluorescent label, it opens up a spectral window for other fluorescent targets increasing multiplexing capabilities. Using membrane stains, DL can be efficiently used to increase the spatial resolution, as we showed by predicting SIM membrane images from a diffraction-limited fluorescence signal using CARE and fnet. The enhanced resolution can improve downstream applications such as analysis of cell cycle stages in spherical bacteria[76]. As bright field or fluorescence membrane images are part of basically any study including microscopy data, we think that artificial labelling and prediction of super-resolution images can be very useful for the bacterial research community.

As a general but important note, DL models are highly specific for the type of data on which they were trained[22]. Evaluating the model on ground truth data is thus essential to validate model performance, identify potential artefacts and avoid a replication crisis for DL-based image analysis[68]. As it is difficult to generate models with good generalisation capabilities, even slightly varying image acquisition parameters can transform a model from a good performer to a source of artefacts. Such parameters include different magnifications (pixel sizes), variations of the focal plane, illumination patterns, camera settings, and many more. However, even if pretrained models do not provide satisfying results, they can be used for transfer-learning, which can strongly accelerate the training and increase model performance. Collecting pre-trained models in model zoos (such as the BioImage model zoo[77]: https://bioimage.io/) can create a database encompassing a variety of species, microscopy techniques and experiments. This database can be used by the researchers to explore potential DL applications and apply pretrained models to their own research using designated platforms[9,14,22,24,43]. Together with easily accessible DL networks and shared datasets (for ours see Supplementary Table 2), this work can support researchers to familiarise themselves with DL and find an entry point into the DL universe.

## Methods

**Segmentation of *E. coli* bright field images.** *E. coli* MG1655 cultures were grown in LB Miller at 37 °C and 220 rounds per minute (rpm) overnight. Working cultures were inoculated 1:200 and grown at 23 °C and 220 rpm to OD600 ~ 0.5–0.8. For time-lapse imaging, cells were immobilised under agarose pads prepared using microarray slides (VWR, catalogue number 732-4826) as described in de Jong et al., 2011[44]. Bright field time series (1 frame/min, 80 min total length) of 10 regions of interest were recorded with an Andor iXon Ultra 897 EMCCD camera (Oxford instruments) attached to a Nikon Eclipse Ti inverted microscope (Nikon Instruments) bearing a motorised XY-stage (Märzhäuser) and an APO TIRF 1.49NA 100x oil objective (Nikon Instruments). To generate the segmentation training data, individual frames from different regions of interest were rescaled using Fiji (2x scaling without interpolation) to allow for better annotation and to match the receptive field of the network. Resulting images were annotated manually using the freehand selection ROI tool in Fiji. For quality control, a test dataset of 15 frames was generated similarly. Contrast was enhanced in Fiji and images were either converted into 8-bit TIFF (CARE, U-Net, StarDist) or PNG format (pix2pix).

To track cells during their release from stationary phase, we immobilised cells from an ON culture as described above. Time series of multiple positions were recorded at 2 min interval (40 min in total). To account for the small size of the cells, we used an additional tube lens (1.5x) to reduce the pixel size to 106 nm. To obtain the training dataset, we recorded stationary phase cells directly after immobilization and additionally annotated selected individual frames from the time series.

**Data pre- and post-processing for cell segmentation using the multi-label U-Net notebook.** In order to improve segmentation performance, we employed a U-Net that is trained on semantic segmentations of both cell cytosol and boundaries. To generate the respective training data, annotated cells were filled with a grey value of 1, while cell boundaries were drawn with a grey value of 2 and a line thickness of 1. Together with the fluorescence image, this image was used as network input during training. During post-processing, cell boundaries were subtracted from predicted cell segmentations, followed by thresholding and marker-based watershed segmentation (Fiji plugin "MorpholibJ")[78]. Pre- and post-processing routines are provided as Fiji macros and can be downloaded from the DeepBacs github repository.

**Segmentation of *S. aureus* bright field and fluorescence images.** For *S. aureus* time-lapse experiments overnight cultures of *S. aureus* strain JE2 were back-diluted 1:500 in tryptic soy broth (TSB) and grown to mid-exponential phase (OD$_{600}$ = 0.5). One millilitre of the culture was incubated for 5 min (at 37 °C) with the membrane dye Nile Red (5 µg/ml, Invitrogen), washed once with phosphate buffered saline (PBS), subsequently pelleted and resuspended in 20 µl PBS. One microlitre of the labelled culture was then placed on a microscope slide covered with a thin layer of agarose (1.2% (w/v) in 1:1 PBS/TSB solution). Time-lapse images were acquired every 25 s (for differential interference contrast (DIC)) and 5 min (for fluorescence images) by structured illumination microscopy (SIM) or classical diffraction limited widefield microscopy in a Deltavision OMX system (with temperature and humidity control, 37 °C). The images were acquired using 2 PCO Edge 5.5 sCMOS cameras (one for DIC, one for fluorescence), an Olympus 60×1.42NA Oil immersion objective (oil refractive index 1.522), Cy3 fluorescence filter sets (for the 561 nm laser) and DIC optics. Each time-point results from a Z-stack of 3 epifluorescence images using either the 3D-SIM optical path (for SIM images) or classical widefield optical path (for non-super-resolution images). These stacks were acquired with a Z step of 125 nm in order to use the 3D-SIM-reconstruction modality (for the SIM images) of Applied Precision's softWorx software (AcquireSRsoftWoRx v7.0.0 release RC6), as this provides higher quality reconstructions. A 561 nm laser (100 mW) was used at 11–18 W cm$^{-2}$ with exposure times of 10–30 ms. For single-acquisition *S. aureus* experiments, sample preparation and image acquisition were performed as mentioned above but single images were acquired. To generate the training dataset for StarDist segmentation, individual channels were separated and pre-processed using Fiji[9,43]. Nile Red fluorescence images were manually annotated using ellipsoid selections to approximate the *S. aureus* cell shape. Resulting ROIs were used to generate the

required ROI map images (using the "ROI map" command included in the Fiji plugin LOCI) in which each individual cell is represented by an area with a unique integer value. Training images ($512 \times 512$ px²) were further split into $256 \times 256$ px² images, resulting in 28 training images pairs. 5 full field-of-view test image pairs were provided for model quality control. For segmentation dataset of *S. aureus* bright field images, we used the ROI masks created for Nile Red fluorescence image segmentation, as both images were acquired in parallel.

**Segmentation of live *B. subtilis* cells.** *B. subtilis* cells expressing FtsZ-GFP (strain SH130, PY79 Δhag ftsZ::ftsZ-gfp-cam) were prepared as described in Whitley et al., 2021[47]. Strains were taken from glycerol stocks kept at −80 °C and streaked onto nutrient agar (NA) plates containing 5 µg/ml chloramphenicol then grown overnight at 37 °C. Liquid cultures were started by inoculating time-lapse medium (TLM) (de Jong et al., 2011)[44] with a single colony and growing overnight at 30 °C with 200 rpm agitation. The following morning, cultures were diluted into chemically defined medium (CDM) containing 5 µg/ml chloramphenicol to $OD_{600} = 0.1$, and grown at 30 °C until the required optical density was achieved[47]. All imaging was done on a custom built, 100X inverted microscope. A 100x TIRF objective (Nikon CFI Apochromat TIRF 100XC Oil), a 200 mm tube lens (Thorlabs TTL200) and Prime BSI sCMOS camera (Teledyne Photometrics) were used achieving an imaging pixel size of 65 nm/pixel. Cells were illuminated with a 488 nm laser (Obis) and imaged using a custom ring-TIRF module operated in ring-HiLO[79]. A pair of galvanometer mirrors (Thorlabs) spinning at 200 Hz provides uniform, high SNR illumination. The raw data analysed here were acquired and analysis of that raw data presented in Whitley et al., 2021[47]. These data have now been reanalysed using cell segmentation methods discussed. Slides were prepared as described previously. Molten 2% agarose made with CDM was poured into gene frames (Thermo Scientific) to form flat agarose pads, then cut down to thin 5 mm strips. 0.5 µl of cell culture grown to mid-exponential phase ($OD_{600} = 0.2$–0.3) was spotted onto the agarose and allowed to absorb (~30 s). A plasma-cleaned coverslip was then placed atop the gene frame and sealed in place. Before imaging, the prepared slides were then pre-warmed inside the microscope body at least 15 min before imaging. Time-lapse images were then taken in TIRF using a custom built 100x inverted microscope. Images were taken at 1 s exposure, 1 frame/min at 1–8 W/cm²[47]. Videos were denoised using ImageJ plugin PureDenoise[35] then lateral drift was corrected using StackReg[80].

To create the training dataset, 10 frames were extracted from each time-lapse ~10 frames apart. This was to ensure sufficient difference between the images used for training. Ground truth segmentation maps were generated by manual annotation of cells in each frame using the Fiji/ImageJ LabKit plugin lab (https://github.com/juglab/imglib2-labkit). This process assigns a distinct integer to all pixels within a cell region, and background pixels are labelled 0. A total of 4,672 cells were labelled across 80 distinct frames to create the final training dataset.

**Confocal imaging for denoising of *E. coli* time series.** *E. coli* strain CS01 carrying a chromosomal H-NS-mScarlet-I protein fusion (parental strain NO34) was grown in LB Lennox at 25 °C and shaking at 220 rpm. To generate the training dataset, cells were fixed chemically using a mixture of 2% formaldehyde and 0.1% glutaraldehyde. Fixed or live cells were immobilised under agarose pads poured into gene frames following the protocol by de Jong et al.[44]. Imaging was performed on a commercial Leica SP8 confocal microscope (Leica Microsystems) bearing a 1.40 NA 63x oil immersion objective (Leica Microsystems). To increase optical sectioning, the pinhole size was set to 0.5 airy units and $512 \times 512$ px² confocal images (45 nm pixel size) were recorded. Emission was detected with HyD detectors in standard operation mode (gain 100, detection window 570–650 nm). For the training dataset, a two-channel image of the same structure was recorded in frame sequential mode using different settings for low (0.03% 561 nm laser light, no averaging) and high SNR images (0.1% 561 nm laser light, 4x line averaging), respectively. For live-cell time series, the field of view was reduced to $256 \times 256$ px² to allow for fast acquisition of high SNR images at ~0.8 Hz. Low SNR time series were recorded at similar frame rate by including a lag time. Similar settings were used for the MreB denoising dataset, except that sfGFP was excited with 488 nm and fluorescence was detected between 495 nm and 560 nm. To increase optical sectioning even further to optimized observation of processive MreB movement, the pinhole size was set to 0.3 airy units.

**B. subtilis VerCINI microscopy.** The raw data analysed here were acquired and analysis of that raw data is presented in Whitley et al. 2021[47]. These data have now been reanalysed using the denoising methods described. Silicone micropillar wafers were nanofabricated and used to prepare agarose microholes as described in Whitley et al., 2021[47]. Molten 6% agarose was poured onto the silicone micropillars and allowed to set, forming an agarose pad punctured with microscopic holes. The agarose pad was then transferred into a gene frame, and agarose surrounding the micro-hole array was cut away. Concentrated liquid cell culture at mid-exponential phase ($OD_{600} = 0.4$) was loaded onto the pad and centrifugation using an Eppendorf 5810 centrifuge with MTP/Flex buckets loaded individual cells into the microholes. The pad was then washed to remove unloaded cells. This repeated several times until a sufficient level cell loading was achieved. Cells were imaged at 1 frame/second with continuous exposure for 2 min at 1–8 W/cm²[47]. Image

denoising was performed using the ImageJ plugin PureDenoise[35] and lateral drift was then corrected using StackReg[80].

**E. coli cell cycle classification.** Classification of rod-shaped, dividing and microcolonies was performed using the time series described in section '*Segmentation of E. coli bright field images*'. Individual frames from several time series were used for training. To generate the training dataset, individual frames spread over the entire time series (typically frames 1, 15, 30, 55 and 80) were converted into PNG format. For the large field-of-view model, the entire image was used, while images were split into 4 regions of $256 \times 256$ px² size for the small field-of-view model. Images were annotated using LabelImg[49]. The final training dataset contained 25 (100 for small field-of-view) annotated patches, and dataset size was increased 4x during training using data augmentation implemented in the ZeroCostDL4Mic YOLOv2 notebook (rotation and flipping).

**E. coli antibiotic phenotyping.** *E. coli* strain NO34[55] was grown in LB at 32 °C shaking at 220 rpm overnight. Working cultures were inoculated 1:200 in fresh LB and grown to mid-exponential phase and antibiotics were added at the concentration and for the time listed in Supplementary Table 7. Antibiotic stock solutions were prepared freshly 5-10 min before use. Cells were fixed using a mixture of 2% formaldehyde and 0.1% glutaraldehyde, quenched using 0.1% sodium borohydrate (w/v) in PBS for 3 min and immobilised on PLL-coated chamber slides (see Spahn et al. 2018 for details)[64]. Nucleoids were stained using 300 nM DAPI for 15 min. After three washes with PBS, 100 nM Nile Red in PBS was added to the chambers and confocal images were recorded with a commercial LSM710 microscope (Zeiss, Germany) bearing a Plan-Apo 63x oil objective (1.4 NA) and using 405 nm (DAPI) and 543 nm (Nile Red) laser excitation in sequential mode. Images ($800 \times 800$ px²) were recorded with a pixel size of 84 nm, 16-bit image depth, 16.2 µs pixel dwell time, 2x line averaging and 1 airy unit pinhole size.

Four to eight confocal images were used to generate the training dataset, depending on the cell count per image (for example, only few cells are present per image for nalidixate-treated cells, while many cells were present for chloramphenicol treatment). Each image was converted to PNG format, split into 4 non-overlapping patches ($400 \times 400$ px²) and patches were annotated online using makesense.ai[81]. Annotations were exported in PASCAL VOC format. Next to the 5 antibiotic treatments and control conditions, vesicles and partially attached cells were added as additional classes ("Vesicles" and "Oblique", respectively), resulting in a total of eight classes. Synthetic test data was generated by randomly stitching $200 \times 200$ px² patches of different drug treatments and the control condition. Small patches were manually cropped from images that were not seen by the network during the training. In total, 32 test images were generated this way and annotated online using makesense.ai[81] as described above. Additionally, $400 \times 400$ px² image patches of previously unseen images (drug treatments and control) were annotated using LabelImg[49].

**Artificial labelling of *E. coli* membranes.** PAINT super-resolution images of *E. coli* membranes were recorded as described elsewhere[64]. In brief, cells were grown in LB at 37 °C and 220 rpm, fixed in mid-exponential phase ($OD_{600} = 0.5$) using a mixture of 2% formaldehyde and 0.1% glutaraldehyde, immobilised on poly-L-Lysine coated chamber slides and permeabilised with 0.5% TX-100 in PBS for 30 min. 400 pM Nile Red in PBS was added and PAINT time series (6,000–10,000 frames) were recorded on a custom built setup for single-molecule detection (Nikon Ti-E body equipped with a 100x Plan Apo TIRF 1.49 NA oil objective) using 561 nm excitation (~1 kW/cm²) or a commercial N-STORM system with a similar objective and imaging parameters. Two image datasets were recorded using either a 1x or 1.5x tube lens (158 and 106 nm pixel size, respectively). PAINT images were reconstructed using Picasso v.0.2.8 and v.0.3.3[82] and exported at different magnifications (8x for 158 nm pixel [19.8 nm/px] and 6x for 106 nm pixel size [17.7 nm/px]). Corresponding bright field images were scaled similarly in Fiji without interpolation and registered with the PAINT image. Multiple $512 \times 512$ px² image patches were extracted from these images and used for model training. For artificial labelling in drug-treated cells, cells were exposed to the following antibiotics: 100 µg/ml rifampicin for 10 min, 50 µg/ml Chloramphenicol for 60 min, 2 µg/ml Mecillinam for 60 min. Further sample preparation and imaging was performed similar to untreated cells.

**Prediction of membrane SIM images in live *E. coli* and *S. aureus* cells.** For widefield-to-SIM prediction experiments overnight cultures of *E. coli* strain DH5α were back-diluted 1:500 in LB and grown to mid-exponential phase ($OD_{600} = 0.3$). One millilitre of the culture was incubated for 10 min (at 37 °C) with the membrane dye FM5-95 (10 µg/ml, Invitrogen), washed once with PBS, subsequently pelleted and resuspended in 10 µl PBS. One microliter of the labelled culture was then placed on a microscope slide covered with a thin layer of agarose (1.2% (w/v) in 1:1 PBS/LB solution). Image acquisition was performed as mentioned in section "Segmentation of *S. aureus* bright field and fluorescence images".

To generate the paired training dataset for super-resolution prediction, raw SIM images were averaged to obtain the diffraction limited widefield image, while the in-focus plane of the SIM reconstruction was used as corresponding high-resolution image. The dataset was curated by removing defocused images and

images with low signal resulting in reconstruction artefacts. In total, 55 training and five test image pairs were used for *E. coli*. For *S. aureus*, this resulted in 94 training and five test image pairs.

**Data augmentation**. As a general strategy to increase training dataset sizes, we used data augmentation[22,83] for all DL learning tasks performed in this study using mostly image rotation and flipping.

**Calculation of the multiscale structural similarity index (SSIM)**. Performance of several DL approaches (e.g. CARE) was accessed by calculating the multiscale structural similarity index (here denoted as SSIM) between the source/predicted image and the ground truth image[54] (see Supplementary Note 1). Since background is suppressed efficiently by most networks and is thus over-proportionally contributing to the average per-image SSIM value (leading to an over-optimistic value), we determined the SSIM only within the outlines of bacterial cells. For this, ROIs were generated in Fiji by thresholding the high SNR image or time series average image. For denoising of live-cell time series lacking ground truth data (e.g. N2V), we determined the SSIM value over time by comparing each image frame to the subsequent image frame of the time series (thus termed subsequent-frame SSIM). A low SSIM value thus depicts a high frame-to-frame variation.

**Tracking analysis using TrackMate**. To track exponentially growing cells (Supplementary Video 3) and cells transitioning from stationary to lag phase (Supplementary Video 4), we used the 'mask image detector' in DL-capable version of TrackMate[46]. No thresholding was used on the detected labels and the LAP tracker was used with 10 px linking distance and segment gap closing (5 px). To track MreB filaments, we used the LoG detector with a radius of 0.25 μm (0.5 μm diameter) and varying thresholds for low SNR, high SNR and denoised time series. Linking distance was set to 0.2 μm using the simple LAP tracker.

**SQUIRREL analysis**. To access artefacts in super-resolution prediction from widefield data we used the SQUIRREL algorithm implemented in the Fiji NanoJ plugin[66,84]. This way, the predictions of 5 WF images and the respective SIM ground truth images were analysed. SQUIRREL calculates a diffraction limited image from super-resolution images to compare them with the corresponding low-resolution ground truth image. Resulting error maps give rise to reconstruction and in this case also prediction artefacts.

**Statistics and reproducibility**. For the majority of datasets, multiple images or time series were recorded in a single imaging session. It was ensured that the acquired data is representative by the different expert laboratories contribution to this work. For object detection (drug-treated cells) and artificial labelling (super-resolution), images from 2–3 independent experiments were included in the training and test dataset. Information about the number of training images and/or cell count per image is provided in the Supplementary Information and the Supplementary Data 1. The latter also includes the individual values used for statistical analysis.

**Reporting summary**. Further information on research design is available in the Nature Research Reporting Summary linked to this article.

## Data availability

Datasets and models generated in this work can be downloaded via Zenodo (**see** Supplementary Table 2), while further documentation on sample preparation, data pre-processing, training parameters and example images can be found on our GitHub repository (https://github.com/HenriquesLab/DeepBacs/wiki).

## Code availability

Notebooks can be accessed via the ZeroCostDL4Mic repository (https://github.com/HenriquesLab/ZeroCostDL4Mic/wiki). Macros for image pre- and postprocessing for the Multi-label U-Net can be found in the DeepBacs repository (https://github.com/HenriquesLab/DeepBacs/wiki).

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

## Acknowledgements

C.S. and M.H. acknowledge funding by the Deutsche Forschungsgemeinschaft (German Science Foundation; DFG), grants HE 6166/17-1 and SFB 1177. C.S. further acknowledges support by the European Molecular Biology Organization (EMBO) in form of a Scientific Exchange Grant (grant nr. 8587). R.F.L. would like to acknowledge the support of the MRC Skills development fellowship (MR/T027924/1). P.P. acknowledges funding by a La Caixa Junior Leader Fellowship (LCF/BQ/PI20/11760012) financed by "la Caixa" Foundation (ID 100010434) and by European Union's Horizon 2020 research and innovation programme under the Marie Skłodowska-Curie grant agreement No 847648. Further, P.P. acknowledges funding by a Science and Technology Foundation grant (PTDC/BIA-MIC/2422/2020) and by Project LISBOA-01-0145-FEDER-007660 Micro-biologia Molecular, Estrutural e Celular (to ITQB-NOVA). M.G.P. acknowledges funding by the European Research Council (ERC-2017-CoG-771709), MOSTMICRO-ITQB R&D Unit (UIDB/04612/2020, UIDP/04612/2020) and LS4FUTURE Associated Laboratory (LA/P/0087/2020). SH acknowledges funding support by a Wellcome Trust & Royal Society Sir Henry Dale Fellowship (206670/Z/17/Z). MC supported by a UK Medical Research Council doctoral studentship. G.J. was supported by grants awarded by the Finnish Cancer Organization, the Sigrid Juselius Foundation, the Academy of Finland (338537), the Åbo Akademi University Research Foundation (CoE CellMech; to G.J.), and the Drug Discovery and Diagnostics strategic funding to Åbo Akademi University. E.G.d.M. and R.H. are supported by Gulbenkian Foundation and received funding from the European Research Council (ERC) under the European Union's Horizon 2020

research and innovation programme (grant agreement No. 101001332) (to R.H.), the European Molecular Biology Organization (EMBO) Installation Grant (EMBO-2020-IG-4734) (R.H.) and the Wellcome Trust (203276/Z/16/Z) (R.H.). R.H. is further supported by a Chan Zuckerberg Initiative Visual Proteomics Grant (vpi-0000000044). The authors thank Alexandre Bisson for sharing *Agrobacterium tumefaciens* live-cell data and Kevin D. Whitley for providing *B. subtilis* FtsZ-GFP data.

## Author contributions

C.S., M.H. and R.H. conceived the project; L.v.C., R.F.L., E.G.d.M., G.J., and R.H. wrote source code in the ZeroCostDL4Mic project; C.S., P.M.P. and M.C. performed the image acquisition of the training and test data; C.S. and M.C. annotated the data. E.G.d.M. helped with model training and data analysis. M.G.P and S.H. provided data and helped writing the manuscript. S.H. further performed MreB tracking analysis. C.S. wrote the manuscript with input from all co-authors.

## Competing interests

The authors declare no competing interests.
