## [Peer Review File · Communications Biology]

Reviewers' comments:

Reviewer #1 (Remarks to the Author):

Summary

The article presents a combined platform called ZeroCostDL4Mic which brings together the DL based automated cell imaging efforts of segmentation, object detection, denoising, artificial labelling and prediction of super-resolution images. The research is adequate with all the codes, data, video, supplementary figures and tables. The reviewer found the suggested web based platform to be interactive and easy to use. The effort from the authors made usage of DL based techniques in microscopy and cell imaging research and made the research easily accessible to new researchers in the field.

Overall Impression

The article is rich and informative, however, there is considerable room for improvement in paper structure. It is difficult to follow the flow of the problem description to solution it addresses. There are several repeat sentences and misplaced contents. For example, items that are suitable for the introduction are written in the discussion. In the result section there is description of an experiment that can go to the Methods section. The open source algorithms that are used should be discussed in the introduction section, the result section is exclusively for reporting results. Several such changes (not exhaustive) are requested in the following section.

The Result and Discussion sections can be made more concise and summarized. Authors presented all findings and insights of the research which were out of scope of the article as described in the abstract. Efficacy of methods depending on cell types and microscopy process can be given in the form of table for better understanding and usage.

Things that are needed to be addressed in the revised version

1. What part of the work is the original work of the authors and what part comes from the open source contribution of the user community? This needs to be clearly delineated.
2. What are the changes that were made to the algorithms and the end result and contribution of the change?

Requests:

1. A short workflow on usage of the platform or graphical abstract explaining ZerocostDL4Mic.
2. List of the algorithms used per task, their applicability as well as strong and weak zones for different cell and image types and computational, hardware and time requirements in the form of a table.

Abstract

Line 1 – 8: Introductory lines, can go in introduction. Authors discussed various aspects of the methods in the results section, but it was not stated in the abstract.

Introduction

Line 69: "...while other DL assisted bio imaging tasks remain largely underexploited." Here detail description of the word "other" is expected. The gap which is addressed is not clearly stated. From recent literatures it is observed that there had been considerable works on detection, de noising, labelling and image quality development. Exactly which part is inadequate which is fulfilled by this research should be stated clearly.

Deep learning based cell image segmentation, classification, denoising, labelling is already performed by other research groups as mentioned from reference 1-27. What advantages does

ZerocostDL4Mic have? Clearly explain. Does it simply package other techniques or does it make improvements on these techniques for better usage?

Result

119 should go to introduction

148 – 162 should go to introduction

163 – 180 should go to methods section

For all other tasks, please summarize the results and send relevant procedures to methods section.

Discussion

631 - 635 This should be in the materials and methods; discussion is for summarizing the task only.

636 - 640 this part is repeated.

640 - 644 This part is not necessary, it is already in the introduction.

Reviewer #2 (Remarks to the Author):

The authors showcase the use of different deep learning-based image analysis approaches to analyse bacterial images. The work is based on the Zero-CostDL4Mic platform (recently developed by the same authors and published in Nature Communications in 2021).

These results can be of interest for both the biological and the machine learning/AI communities. The paper is overall well structured and written, figures are clear, methodologies are well reported, and code is available to the community.

I have however some general comments about the manuscript, which the authors should carefully consider:

- 1) the platform has already been published, but in von Chamier et al 2021 has been used for mammalian cells; as the type of analysis is basically the same (although performed here for bacterial cells), the authors should carefully discuss differences/similarities with their previous work, so to point the reader to cell type-specific issues, and to clarify the novelty of this work;
- 2) the authors should discuss issues with tracking in case of touching objects (or even better try to implement tracking in such a case);
- 3) it might be worth considering adding a table/box where the various models/algorithms mentioned are defined and briefly described to help a non-expert reader;
- 4) it is not obvious how results and conclusions depend on the specific datasets considered, the authors should at least comment on this point.

DeepBacs – Point-to-point response

We thank the reviewers for their time and dedication in evaluating our manuscript, as well as for their excellent comments. We added additional content to the manuscript and restructured large sections based on the input of the reviewer, which made the manuscript clearer. Adding general descriptions of the networks, as well as summarizing their advantages/disadvantages in a table, supports the scope of the study to educate microbiologists how to use deep learning approaches for different tasks. We also added new data showing tracking with StarDist and TrackMate, as well as a challenging denoising dataset to illustrate caveats of deep learning. We think that the changes improved the manuscript significantly and hope that they address the questions raised by the reviewers.

To implement boxes and to simplify manuscript revision, we converted the manuscript from our overleaf format into a word document. As this led to changes in line numbers and arrangement, we are not able to refer to the original numbering and apologise for this inconvenience. However, we provided a document in which we highlighted the main changes made during the revision.

Reviewer #1 (Remarks to the Author):

Summary

The article presents a combined platform called ZeroCostDL4Mic which brings together the DL based automated cell imaging efforts of segmentation, object detection, denoising, artificial labelling and prediction of super-resolution images. The research is adequate with all the codes, data, video, supplementary figures and tables. The reviewer found the suggested web-based platform to be interactive and easy to use. The effort from the authors made usage of DL based techniques in microscopy and cell imaging research and made the research easily accessible to new researchers in the field.

Response: We thank the reviewer for their positive evaluation of our study. Previous implementations of the ZeroCostDL4Mic platform focused on the analysis of mammalian cells. The work we here present provides a new view into strategies for using these tools successfully also for bacterial microscopy images. In addition, we also extended the platform by adding the multi-class U-Net, particular useful in bacterial segmentation. We are convinced that our work can inspire bacteriologists and provide them with excellent tools to start adapting deep learning into their imaging. We now stated this more clearly in the text.

Overall Impression

The article is rich and informative, however, there is considerable room for improvement in paper structure. It is difficult to follow the flow of the problem description to solution it addresses. There are several repeat sentences and misplaced contents. For example, items that are suitable for the introduction are written in the discussion. In the result section there is description of an experiment that can go to the Methods section. The open source algorithms that are used should be discussed in the introduction section, the result section is exclusively for reporting results. Several such changes (not exhaustive) are requested in the following section.

Response: We agree with the reviewer and thank them for pointing out the misplaced content. We rearranged sections and hope that this improved the structure of the manuscript. To provide an overview to non-expert readers, we also added boxes that explain the different networks, as well as the metrics used in this study.

The Result and Discussion sections can be made more concise and summarized. Authors presented all findings and insights of the research which were out of scope of the article as described in the abstract. Efficacy of methods depending on cell types and microscopy process can be given in the form of table for better understanding and usage.

Response: We now moved large sections from the results table into the introduction. As suggested by the reviewer, we summarized the performances of the different models and tasks in Table 1. Details about the datasets, as well as model training and performance, are already provided in the Supplementary information and were thus not added to the main text.

Things that are needed to be addressed in the revised version

1. What part of the work is the original work of the authors and what part comes from the open source contribution of the user community? This needs to be clearly delineated.

Response: As described above, we employed existing networks that are integrated into the ZeroCostDL4Mic platform. Except for the multi-label network, all networks were already available. Our intention was not to create novel code, but to demonstrate how existing networks can be used specifically for bacterial datasets. We now rephrased parts of the manuscript to make this clearer.

2. What are the changes that were made to the algorithms and the end result and contribution of the change?

Response: Only minor changes to the code were made within this work. The main part of this work is leveraging existing code by optimizing hyperparameters and providing material (datasets, models, settings and experiences) to educate novel users in the application of deep learning.

Requests:

1. A short workflow on usage of the platform or graphical abstract explaining ZeroCostDL4Mic.

Response: The authors thank the reviewer for this great suggestion. We now added a graphical abstract to the manuscript.

2. List of the algorithms used per task, their applicability as well as strong and weak zones for different cell and image types and computational, hardware and time requirements in the form of a table.

Response: We have now added a table that gives an overview on the networks used for specific tasks, their strong and weak zones, and for which cell type we would recommend them (Table 2). We did not add hardware and time requirements, as we solely employed the networks in the cloud-based ZeroCostDL4Mic platform. As training times vary depending on the hyperparameters and dataset size, we did specify them but classified it into slow, intermediate and fast training speed. Specific parameters for the networks trained in this work are provided in the Supplementary Tables.

Abstract

Line 1 – 8: Introductory lines, can go in introduction. Authors discussed various aspects of the methods in the results section, but it was not stated in the abstract.

Response: We removed the first lines from the abstract.

Introduction

Line 69: "...while other DL assisted bio imaging tasks remain largely underexploited." Here detail description of the word "other" is expected. The gap which is addressed is not clearly stated. From recent literatures it is observed that there had been considerable works on detection, de noising, labelling and image quality development. Exactly which part is inadequate which is fulfilled by this research should be stated clearly.

Deep learning based cell image segmentation, classification, denoising, labelling is already performed by other research groups as mentioned from reference 1-27. What advantages does ZerocostDL4Mic have? Clearly explain. Does it simply package other techniques or does it make improvements on these techniques for better usage?

Response: We thank the reviewer to pointing out the generic character of this sentence. It is true that the mentioned tasks were successfully performed before. However, this is mainly the case for eukaryotic specimen such as individual cells or small organisms. Although these methods are widely applicable, we did not find much application in bacteriology and speculated that the technology is not well known in this field. As popularity in the community increases with the number of studies demonstrating the potential for their model organism or question, we aimed to provide meaningful examples for DL in microbiology. As ZeroCostDL4Mic streamlines available models, and provides further possibilities such as transfer learning or model quality assessment, it is well suited for novice users to test and gain experience with DL. The standardized format of the notebooks, in our opinion, makes the networks straight-forward to use, which is also reflected by the positive response of the community. Again, we want to emphasize that our main goal is to educate researchers on how to use DL in bacteriology and to demonstrate the potential of this technology by providing illustrative and meaningful examples.

Result

119 should go to introduction

Response: Moved accordingly

148 – 162 should go to introduction

Response: As we showcase various tasks, our introduction concentrated on the general aspect of deep learning at its deployment in image analysis. Although we now moved the descriptions of the individual tasks to the introduction as suggested by the reviewers, we feel that this particular section is too detailed and fits better to where it is. We thus did not move this particular section.

163 – 180 should go to methods section

Response: Moved accordingly

For all other tasks, please summarize the results and send relevant procedures to methods section.

Response: We moved large sections to either introduction or methods and summarized the results.

Discussion

631 - 635 This should be in the materials and methods; discussion is for summarizing the task only.

636 - 640 this part is repeated.

640 - 644 This part is not necessary, it is already in the introduction.

Response: The authors thank the reviewer for these suggestions. The sections were changed accordingly. Also, we restructured the discussion into sections for each task, synthesizing the results and highlighted general aspects of the different networks.

Reviewer #2 (Remarks to the Author):

The authors showcase the use of different deep learning-based image analysis approaches to analyse bacterial images. The work is based on the Zero-CostDL4Mic platform (recently developed by the same authors and published in Nature Communications in 2021).

These results can be of interest for both the biological and the machine learning/AI communities.

The paper is overall well structured and written, figures are clear, methodologies are well reported, and code is available to the community.

I have however some general comments about the manuscript, which the authors should carefully consider:

1) the platform has already been published, but in von Chamier et al 2021 has been used for mammalian cells; as the type of analysis is basically the same (although performed here for bacterial cells), the authors should carefully discuss differences/similarities with their previous work, so to point the reader to cell type-specific issues, and to clarify the novelty of this work;

Response: We thank the reviewers for the positive feedback on our manuscript. He/she is correct that we mainly use the notebooks provided by the ZeroCostDL4Mic platform. However, the focus of this study was not to add or change a lot of code, but rather to demonstrate the versatility of DL for bacteriology and to educate non-expert readers in using this technology for their purpose. For this we provided datasets, analysis strategies and models in a variety that has, to our knowledge, not been reported before. Although not focusing on novel code, we added an additional notebook (multilabel U-Net) to the ZeroCostDL4Mic platform. We tried to emphasize these aspects in the revised manuscript.

2) the authors should discuss issues with tracking in case of touching objects (or even better try to implement tracking in such a case);

Response: The authors thank the reviewer for this suggestion. We now added a dataset in which we track bacterial cells during recovery from stationary phase using the Fiji plugin TrackMate. We have now added an extra section about tracking to the discussion, in which we briefly discuss tracking at high cell densities.

3) it might be worth considering adding a table/box where the various models/algorithms mentioned are defined and briefly described to help a non-expert reader;

Response: The authors thank the reviewer for this excellent suggestion. We now added two boxes to the manuscript that describe the individual networks and metrics used in this study.

4)it is not obvious how results and conclusions depend on the specific datasets considered, the authors should at least comment on this point.

Response: Based on a suggestion of reviewer #1, we now added a table with strong and weak zones for the networks that we used in our study. This table includes suggestions about which kind of data is suitable for the different networks. In addition, we added a new dataset for denoising, in which we specifically tested the limits of deep learning approaches. For other datasets and tasks, we already included suggestions, for example the size bias for object detection networks or the dependence of segmentation networks on image type and cell morphology.

REVIEWERS' COMMENTS:

Reviewer #1 (Remarks to the Author):

Revisions are complete. Publish.

Reviewer #2 (Remarks to the Author):

The authors have done an excellent job in addressing my concerns, the manuscript is now more clear and better structured.